# Activity-dependent tuning of intrinsic excitability in mouse and human neurogliaform cells

**Ramesh Chittajallu[1]\*, Kurt Auville[1], Vivek Mahadevan[1], Mandy Lai[1], Steven Hunt[1], Daniela Calvigioni[1], Kenneth A Pelkey[1], Kareem A Zaghloul[2], Chris J McBain[1]\***

[1]Laboratory of Cellular and Synaptic Physiology, Eunice Kennedy Shriver National Institute of Child Health and Human Development, National Institutes of Health, Bethesda, United States; [2]Surgical Neurology Branch, National Institutes of Neurological Disorders and Stroke, National Institutes of Health, Bethesda, United States

**Abstract** The ability to modulate the efficacy of synaptic communication between neurons constitutes an essential property critical for normal brain function. Animal models have proved invaluable in revealing a wealth of diverse cellular mechanisms underlying varied plasticity modes. However, to what extent these processes are mirrored in humans is largely uncharted thus questioning their relevance in human circuit function. In this study, we focus on neurogliaform cells, that possess specialized physiological features enabling them to impart a widespread inhibitory influence on neural activity. We demonstrate that this prominent neuronal subtype, embedded in both mouse and human neural circuits, undergo remarkably similar activity-dependent modulation manifesting as epochs of enhanced intrinsic excitability. In principle, these evolutionary conserved plasticity routes likely tune the extent of neurogliaform cell mediated inhibition thus constituting canonical circuit mechanisms underlying human cognitive processing and behavior.

**\*For correspondence:**
ramesh.chittajallu@nih.gov (RC);
mcbainc@mail.nih.gov (CJMB)

**Competing interests:** The authors declare that no competing interests exist.

## Introduction

Although heavily outnumbered by glutamatergic pyramidal cells, the population of diverse GABAergic interneuron (INs) subtypes encompass essential components of cortical and hippocampal circuits (*Kepecs and Fishell, 2014*; *Pelkey et al., 2017*). Appropriate interaction between these broad groups of neurons dictates the overall excitation/inhibition balance critical for normal brain function and its protracted dysregulation can precipitate a wide range of neurological disorders (*Marín, 2012*). The dynamic nature and capacity of this interplay to undergo modifications in the face of ongoing activity is vital for appropriate circuit wiring during development, homeostasis and various cognitive tasks such as learning and memory (*Butt et al., 2017*; *Debanne et al., 2019*; *Keck et al., 2017*; *Pozo and Goda, 2010*; *Takesian and Hensch, 2013*; *Tien and Kerschensteiner, 2018*).

Since the original theoretical postulations by Hebb and early experimental descriptions (*Bliss and Collingridge, 2019*; *Hebb, 1949*), a substantial research effort over many decades has uncovered a bewildering array of pathways/mechanisms for the induction and expression of circuit plasticity (*Debanne et al., 2019*; *Kullmann et al., 2012*; *Malenka and Bear, 2004*). Much of our knowledge concerning the varied routes of plasticity expressed by neuronal subtypes is indebted to animal models. Surprisingly, the extent to which these mechanisms are paralleled in the human central nervous system remain relatively understudied (*Mansvelder et al., 2019*). In particular, the absence of information regarding short- or long-term modulation of excitatory recruitment and output of human INs is striking (*Rózsa et al., 2017*; *Szegedi et al., 2016*).

Successful translation of basic science research conducted in experimental animal models to the clinical setting depends on an integrated knowledge of the similarities and differences in physiological and synaptic properties of neurons in human circuits. In recent years there has been a growing drive and appreciation for such comparisons at the molecular, cellular and physiological levels across evolution (*Beaulieu-Laroche et al., 2018*; *Boldog et al., 2018*; *Hodge et al., 2019*; *Krienen et al., 2019*; *Obermayer et al., 2018*; *Szegedi et al., 2020*; *Eyal et al., 2016*; *Molnár et al., 2016*; *Poorthuis et al., 2018*; *Povysheva et al., 2008*; *Szabadics et al., 2006*; *Varga et al., 2015*). We take advantage in availability of human cortical tissue surgically resected for treatment of patients with drug resistant epilepsy. In every participant, our slice electrophysiological analyses were derived from cortical specimens that lay outside of the seizure onset zone as determined by intracranial recordings and thus deemed electrographically normal (see Materials and methods for details). We focus on neurogliaform cells (NGFCs) that together form a large subset of INs that impart inhibition on distal dendritic domains of cortical and hippocampal pyramidal cells (*Bezaire and Soltesz, 2013*; *Overstreet-Wadiche and McBain, 2015*). Additionally, NGFCs possess the ability to influence other IN subtypes including each other via chemical (GABA mediated) and electrical (gap junction mediated) modalities (*Armstrong et al., 2011*; *Boldog et al., 2018*; *Chittajallu et al., 2013*; *Chu et al., 2003*; *Oláh et al., 2007*; *Price et al., 2005*; *Simon et al., 2005*). Interestingly, NGFC physiology has been studied not only in rodents but also in non-human primates and humans revealing both similar and divergent properties across these species (*Boldog et al., 2018*; *Oláh et al., 2007*; *Poorthuis et al., 2018*; *Povysheva et al., 2007*; *Zaitsev et al., 2009*). However, only a few reports have described short and long-term plasticity impacting recruitment and activity of mouse NGFCs (*Li et al., 2014*; *Mercier et al., 2019*) with a conspicuous lack of published evidence demonstrating any such mechanisms in their human counterparts (*Rózsa et al., 2017*).

In the current study we demonstrate that both mouse and human NGFCs express analogous forms of plasticity via modulation of their intrinsic properties. Comparative studies such as the one described here are vital in determining to what extent circuit features gleaned from experimental animal models are relevant in humans. Particularly, we uncover a previously undescribed evolutionary conserved mechanism that manifests as a short-term enhancement in the propensity of depolarizing inputs to evoke action potential output. Remarkably, amongst the varied IN located in superficial regions of cortical and hippocampal microcircuits, NGFCs in both species were found to be privileged with regard expression of these forms of plasticity. Together our data reveal the presence of cellular mechanisms that result in modulation of intrinsic excitability of NGFCs that represent circuit motifs important for human brain function.

## Results

### Activity-dependent enhancement in intrinsic excitability of NGFCs manifested by barrage firing is evolutionary conserved

In our initial experiments EGFP-positive interneurons (INs) with soma located in the superficial portion of *stratum lacunosum-moleculare* (SLM) of mid-ventral hippocampi and Layer I (LI) of cortex in P35-P60 *Gad2*-EGFP or *Htr3a*-EGFP mice were targeted (see Materials and methods for details; *Chittajallu et al., 2017*; *Lee et al., 2010*; *Tricoire et al., 2011*). Using the approach our focus was restricted to NGFCs derived from progenitors located in the caudal ganglionic eminence (i.e. CGE-NGFCs) (*Overstreet-Wadiche and McBain, 2015*). Approximately 80% of targeted CGE-NGFCs (124 out of 150 EGFP-positive INs) displayed late-spiking (LS; with varying degrees of latency) phenotype and an axo-dendritic morphology (38 EGFP-positive INs) characteristic of NGFCs (*Overstreet-Wadiche and McBain, 2015*; *Figure 1A*; *Figure 1—figure supplement 1A*). Using brief (2 ms) supra-threshold depolarizing current injection steps (1000–1200 pA) we elicited 10 action potentials at a frequency of 30 Hz repeated at 1 s intervals (*Figure 1B*). These conditioning APs resulted in the previously described phenomenon of barrage/persistent firing (BF) (*Deemyad et al., 2018*; *Elgueta et al., 2015*; *Imbrosci et al., 2015*; *Jung and Hoffman, 2009*; *Krook-Magnuson et al., 2011*; *Rózsa et al., 2017*; *Sheffield et al., 2011*; *Suzuki et al., 2014*) in approximately 70% of NGFCs tested (86 out of 124 NGFCs; *Figure 1B*; *Figure 1—figure supplement 1C*). BF constitutes a specific mode of intrinsic plasticity and is characterized by spontaneously generated action potential output in the absence of a depolarizing stimulus. In parallel, INs with

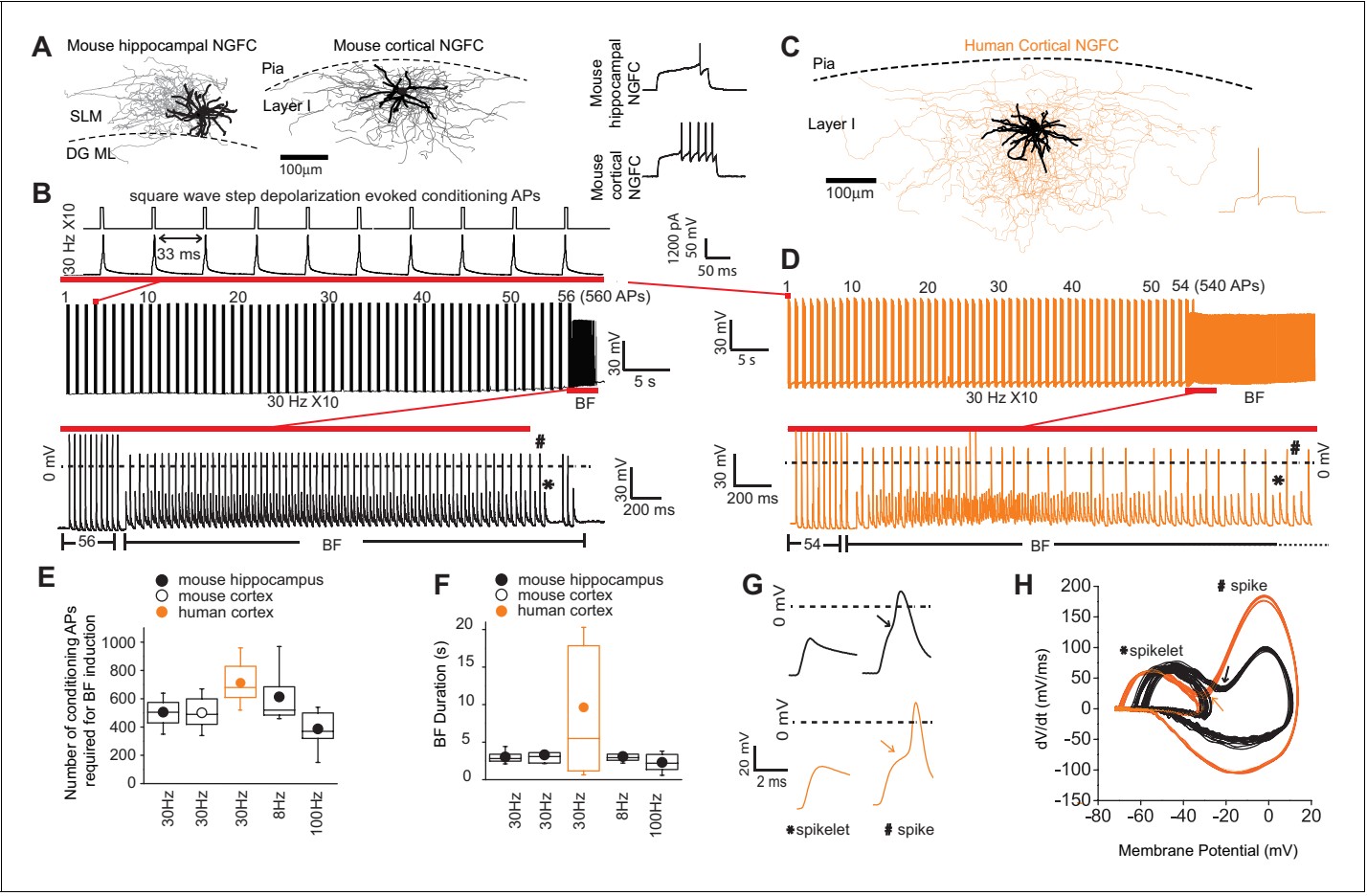

**Figure 1.** Barrage firing (BF) is evolutionary conserved in human NGFCs. (**A,C**) Biocytin reconstruction of mouse (dendrites in black; axon in gray) and human NGFCs (dendrites in black; axon in orange) located in superficial regions of hippocampus (*stratum lacunosum moleculare*; SLM) and cortex (Layer I). (**B,D**) Single example trace showing the induction of barrage firing (BF) in mouse and human NGFCs upon delivery of action potentials via 1 ms depolarizing steps (typically 1000–1200 pA @ 30 Hz and 10 action potentials every second). (**E,F**) Box plots of pooled data depicting the minimum number of action potentials required for emergence of BF and its duration under differing frequencies of inducing action potentials (for the box plots, left to right, the n values are 68, 15, 12, 12 and 7, respectively) (**G,H**) Single example waveforms and corresponding phase plots of spikelets nested within full spikes (the latter defined as those overshooting 0 mV) observed during BF episodes in a single mouse and human NGFC. Black and orange data represent that derived from mouse and human NGFCs, respectively.

The online version of this article includes the following source data and figure supplement(s) for figure 1:

**Source data 1.** Data plotted in *Figure 1*.
**Figure supplement 1.** Privileged induction of barrage firing in mouse and human NGFCs.
**Figure supplement 2.** Induction of multiple barrage firing episodes in individual mouse and human NGFCs.
**Figure supplement 2—source data 1.** Data plotted in *Figure 1—figure supplement 2*.

soma located in layer I (LI) slices of human cortical tissue (resected from patients with pharmaco-resistant epilepsy in order to gain access to remove the hippocampus, the latter structure deemed the focal pathological region; see Materials and methods for details) were also targeted with a sub-population demonstrating a late-spiking pattern (25 cells) identifying them as NGFCs (with additional morphological confirmation in 17 of these 25 cells; *Figure 1C*). Upon delivering the same induction protocol as described above, BF was elicited in 48% of human NGFCs (12/25 NGFCs tested; *Figure 1D*; *Figure 1—figure supplement 1A,C*). Interestingly, even with the number of conditioning APs far exceeding that required to induce BF in NGFCs (800–1500 APs; 30 Hz), BF could not be elicited in superficial cortical and hippocampal INs that were not NGFCs (non-NGFCs), including the recently described human rosehip cell (*Boldog et al., 2018*) (26 and 37 non-NGFCs tested for BF induction in mouse and human, respectively; *Figure 1—figure supplement 1B,D*). We tested

induction protocols consisting of differing frequencies (8 Hz and 100 Hz) and these were also found to elicit BF with essentially similar induction requirements and duration (*Figure 1E,F*). Despite a similar number of conditioning APs necessary for BF induction being similar in both species, the mean BF duration in human NGFCs albeit more variable, was approximately double on average (*Figure 1E,F*). In some recordings, the induction protocol was continued immediately following cessation of BF to ascertain whether subsequent BF epochs of a longer duration could be triggered. However, NGFCs could not display another bout of BF using this experimental approach (*Figure 1—figure supplement 2A*). Interestingly, subsequent BF episodes could be elicited provided a sufficient refractory period had elapsed between delivery of subsequent induction protocols (*Figure 1—figure supplement 2B*). Circumventing such constraints in BF induction (i.e. allowing greater than 1 min before subsequent induction delivery; *Figure 1—figure supplement 2C*) multiple BF episodes (up to four were tested) could be reliably triggered (*Figure 1—figure supplement 2B,D,E*).

Spiking events during NGFC BF episodes in mouse and human were comprised of both spikelets and overshooting action potential spikes (the latter defined as those in which the peak overshot 0 mV; *Figure 1B,D,G*). The spikelet waveform was typically nested within that of the spike as evidenced by a prominent inflection point in the voltage rise of the latter and in corresponding phase plots (*Figure 1G,H*). This biphasic rise, noted in previous studies (*Elgueta et al., 2015*; *Sheffield et al., 2011*), has led to the notion that BF consists of spontaneous AP originating at very distal axonal site(s) that travel retrogradely towards the soma and in some instances subsequently trigger an AP at more proximal sites (*Elgueta et al., 2015*; *Imbrosci et al., 2015*; *Sheffield et al., 2011*) (hence also referred to as retro-axonal BF *Sheffield et al., 2013*). Taken together, these data demonstrate that, of the IN subtypes residing in superficial lamina of cortical and hippocampal microcircuits, NGFCs are privileged with regard an ability to express BF in brief, repeatable episodes. Furthermore, this activity-dependent functional plasticity originating in the distal axonal compartments, is clearly conserved in NGFCs embedded in human cortical circuits.

## Potentiation of responses to proximal depolarization in mouse and human NGFCs

During routine monitoring of recording stability before and immediately after BF cessation, we noted that small depolarizations delivered at the soma employed in voltage clamp to measure access resistance, in some cases, was sufficient in activating action currents. To investigate further, action potential output to depolarizations via a series of 1 s positive current injections before and after BF induction was examined (0 pA to 225 pA; 25 pA steps; *Figure 2A*). A leftward shift in the input/output curve and increased total number of action potentials were evident after excursions into the BF mode (*Figure 2B–C*). The increased excitability observed here occurred in the absence of changes in input resistance (208 ± 53 mΩ and 220 ± 47 mΩ for baseline vs. STP-SE, respectively; n = 11 mouse NGFCs; p=0.33).

To ascertain the duration of this increased excitability, we reverted to voltage-clamp for more precise control of both the holding membrane potential and the depolarizing stimulus. Action currents in any given NGFC were monitored over time following BF cessation in response to a constant test depolarizing voltage command that was found to be ineffective in eliciting any action currents prior to BF induction (+30 mV from a holding potential of −70 mV; *Figure 2D,E*). Using this approach, the potentiated NGFC output following cessation of BF1 was found to persist for a mean duration of approximately 60 s (*Figure 2F,G*). Interestingly, even though the duration of BFs expressed in individual NGFCs does not increase with repeated induction (*Figure 1—figure supplement 2*), the potentiation of somatic depolarization-induced output after the cessation of these subsequent BF episodes was markedly prolonged thus demonstrating the ability to prime this mode of plasticity by multiple rounds of induction (*Figure 2G*; Note: in three instances after cessation of BF4 heightened action potential output lasted 900 s at which point the experiment was terminated).

These data demonstrate that, in addition to the ability to undergo brief periods of distal axonal plasticity (i.e. BF), both mouse and human NGFCs also exhibit an activity-dependent increase in the sensitivity to proximal somatic depolarizing stimuli. For the remainder of the manuscript this mode of plasticity will be termed short-term potentiation of somatic-depolarization driven excitability (STP-SE).

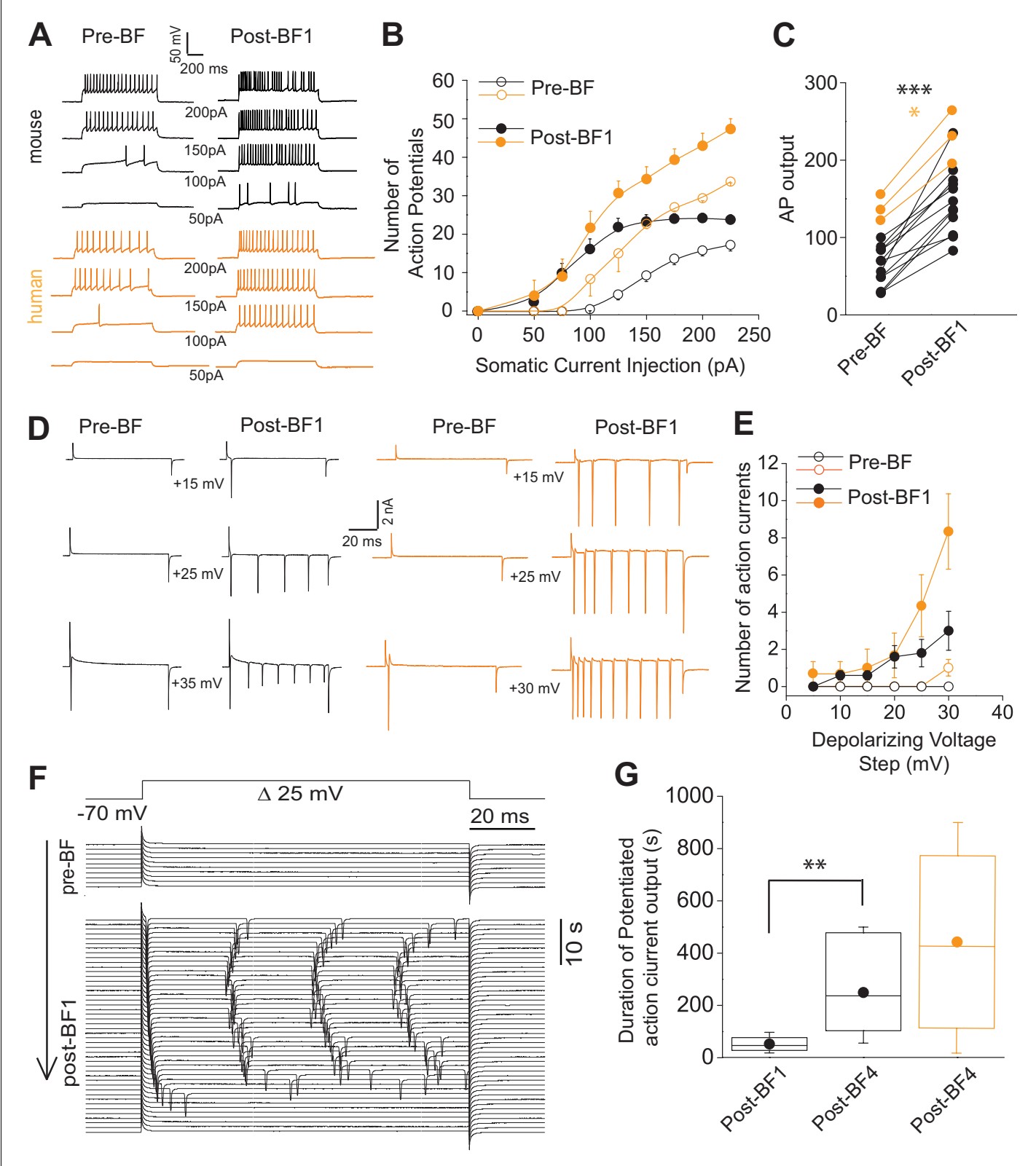

**Figure 2.** Short-term potentiation of somatic-depolarization driven excitability (STP-SE) in mouse and human NGFCs after cessation of BF. (**A**) Single example voltage traces in response to increasing square wave somatic depolarizing current injections before and after induction of BF. Values under each pair of traces (Pre-BF and Post-BF) indicate the size of the 1 s duration current injection. (**B**) Pooled data input/output curves of action potential number in response to increasing amplitudes of depolarizing current injections (50 pA – 220 pA; 25 pA increments; n = 12 and 3 for mouse and human

*Figure 2 continued on next page*

*Figure 2 continued*

NGFCs, respectively). Mouse data are comprised of 8 hippocampal and four cortical NGFCs (**C**) Line plots of AP output before and after BF (calculated as the total sum of APs in response to all current injection steps) in individual mouse (n = 12; paired t-test; p=0.0000083) and human NGFCs (n = 3; paired t-test; p=0.012) tested. (**D**) Single example current traces from NGFCs in response to depolarizing voltage steps (magnitude of steps from a holding potential of −70 mV are indicated under each pair of traces) showing action current output before and after BF. (**E**) Pooled data showing input/output curves depicting number of action currents in response to depolarizing voltage steps (5 mV to 30 mV; 5 mV increments) before and after BF induction (n = 5 and 3 for mouse and human NGFCs, respectively). Mouse data are comprised of 3 hippocampal and two cortical NGFCs (**F**) Single example time course of action current output in response to 25 mV depolarizing voltage steps (sweep interval 10 s) under baseline conditions and after BF induction. (**G**) Box plots depicting the duration of increased action current output to 25 mV depolarizing voltage steps after a single BF episode (post-BF1) and after 4 BF episodes (post BF-4) in mouse (n = 11 and 4 for mouse and human NGFCs, respectively; paired t-test, p=0.007). Mouse data are comprised of 8 hippocampal and three cortical NGFCs.

The online version of this article includes the following source data for figure 2:

**Source data 1.** Data plotted for *Figure 2*.

## Relatively modest activity induces STP-SE independently of BF in both mouse and human NGFCs

Short periods of BF generated in distal axonal region(s) with subsequent driving of action potentials that likely back-propagate into the dendritic tree (*Li et al., 2014*) could conceivably trigger cellular pathways to influence intrinsic excitability resulting in STP-SE. Therefore, we sought to determine whether BF expression is a prerequisite for STP-SE by employing a protocol designed to assay its minimum induction requirements. This was achieved by eliciting action potential spiking in response to test rheobase current injection steps of 250 to 300 ms length via the patch pipette (typically 200 pA). After a baseline period, an induction paradigm was initiated by delivery of conditioning APs as previously described (10 action potentials at 30 Hz) while monitoring action potential output to the same test current pulse (*Figure 3A*). The induction protocol was maintained until commencement of a gradual loss of delay to spiking was noted at which point the conditioning APs were terminated and action potential output to the rheobase current injection continued to be assayed (*Figure 3A*). Notably, with this induction strategy, STP-SE was readily elicited in the absence of measurable BF in mouse hippocampal (n = 9) and human cortical NGFCs (n-4; *Figure 3A–D*) with the mean number of conditioning APs to elicit potentiation in excitability significantly lower than that required to induce BF (*Figure 3E*). As with BF, a subset of NGFCs (7/16 and 2/6 mouse hippocampal and human cortical NGFCs, respectively) and non-late spiking INs (4/4 and 3/3 mouse and human non-NGFCs, respectively) were unable to undergo STP-SE even under protracted period of conditioning APs (*Figure 3D*; *Figure 3—figure supplement 1*). Finally, this potentiation of intrinsic excitability was characterized by a significant hyperpolarization of action potential threshold (*Figure 3F–H*) and reduction in the delay to spiking (*Figure 3B,C,I*) which in combination likely contributes to STP-SE. Together, these data demonstrate the propensity of both mouse and human NGFCs to undergo an activity-dependent potentiation of their action potential output in response to somatic depolarizations without the requirement of entering into BF. For the remainder of the study, we focused our analyses to mouse hippocampal NGFCs to permit insight into the underlying cellular underpinnings for STP-SE expression.

## A transient K$^+$-conductance controls intrinsic excitability of mouse NGFCs

Numerous cellular conductances operating at subthreshold voltages, including those carried by K$^+$-channels, are well suited to influence both the delay to spiking and action potential threshold in a variety of neuronal subtypes. Thus, dynamic modulation of their function constitutes a potentially rapid and powerful cellular mechanism to modulate intrinsic excitability. However, little is known about the complement of K$^+$-channels in positively identified NGFCs (*Overstreet-Wadiche and McBain, 2015*). Under conditions of voltage-gated Na$^+$- and low threshold Ca$^{2+}$-channel block, a depolarizing ramp, indicative of an NGFC identity, is clearly apparent in a subpopulation of *Htr3a*-EGFP SLM interneurons in response to somatic current injections (*Figure 4A*; note for these experiments this parameter was solely employed to define NGFC identity without further morphological confirmation). Restricting our analyses to this subset of recorded neurons, standard voltage-clamp protocols in combination with a pharmacological approach were performed to investigate K$^+$-

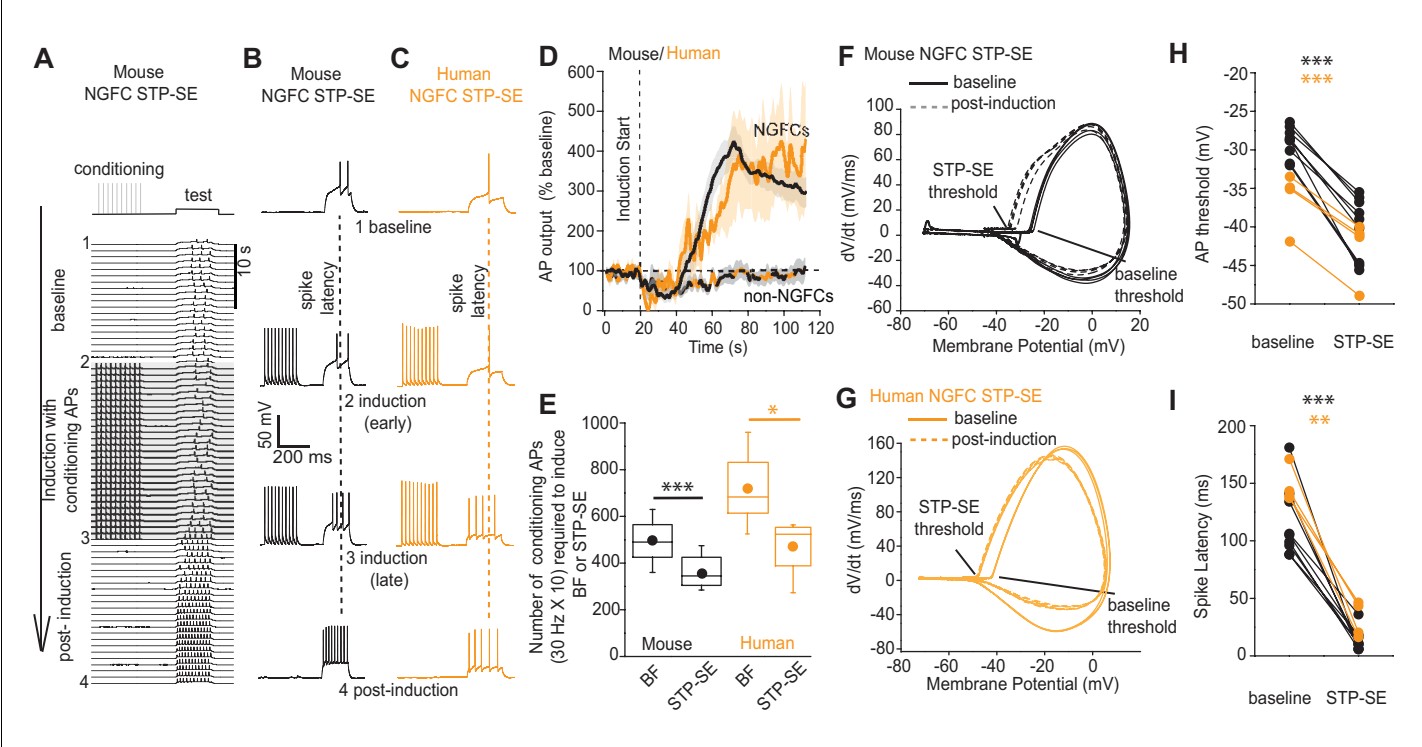

**Figure 3.** STP-SE in mouse and human NGFCs occurs in the absence of BF. (A–C) Single example time course of the conditioning protocol employed to elicit STP-SE in mouse and human NGFCs. Action potential output was elicited to a test rheobase depolarizing current injection (300 ms; typically 100 pA - 140 pA; baseline) in the absence (baseline) and during conditioning action potentials (induction; 10 depolarizing steps of 2 ms width; 1000 pA - 1200 pA). Conditioning action potentials were terminated upon visualization of a marked loss in 1$^{st}$ spike latency (doted vertical lines) invariably accompanied by increased action potential output in response to the test depolarizing current injection (post-induction). (D) Pooled time course of action potential output (measured as number of action potentials in response to the test rheobase current injection) during the entire STP-SE induction protocol. Vertical dotted line indicates the commencement of induction with its termination occurring at varying times for each tested cell. Time course plots are from mouse (black) and human (orange) NGFCs displaying STP-SE (n = 9 and 4 for mouse hippocampal and human cortical NGFCs, respectively). For NGFCs (n = 7 and 2 for mouse hippocampal and human cortical NGFCs, respectively) and non-NGFCs (n = 4 and 3 for mouse and human non-NGFCs, respectively) that did not show STP-SE data were pooled between species (dashed black and orange line). (E) Box plots comparing number of conditioning action potentials required for BF (note these data are replotted from *Figure 1E*; n = 68 and 12, for mouse hippocampal and human cortical NGFCs, respectively) and STP-SE (n = 9 and 4, for mouse hippocampal and human NGFCs, respectively). Paired t-test p-values were 0.0004 and 0.031 for mouse and human, respectively. (F,G) Phase plots of action potential output to the test depolarizing test pulse during baseline (solid lines) and after induction of STP-SE (dotted lines) in single example mouse and human NGFCs. (H,I) Line plots demonstrating hyperpolarization of action potential threshold and reduction of spike latency following induction of STP-SE in mouse hippocampal and human NGFCs. Paired t-test p-values for comparisons of mouse AP threshold and latency (baseline vs. STP-SE) were 0.0000028 and 0.000025, respectively (n = 9). Paired t-test p-values for comparisons of human AP threshold and latency (baseline vs. STP-SE) were 0.000085 and 0.003, respectively (n = 4).

The online version of this article includes the following source data and figure supplement(s) for figure 3:

**Source data 1.** Data plotted for *Figure 3*.

**Figure supplement 1.** Inability to induce STP-SE in a subpopulation of NGFCs and all non-NGFCs tested.

channel expression. Depolarizing steps from −70 mV to membrane potentials close to AP threshold (AP threshold in NGFCs = −29 ± 0.3 mV; n = 70) revealed both inactivating and to a lesser extent sustained K$^+$-conductances operating at subthreshold membrane potentials (*Figure 4B,C*). Activity of subthreshold Kv1 conductances, in particular Kv1.1 (often referred to as D-type) have been linked to the phenomenon of late-spiking in excitatory and inhibitory neuronal subtypes (*Bekkers and Delaney, 2001*; *Campanac et al., 2013*; *Dehorter et al., 2015*; *Storm, 1988*). However, the sub-threshold K$^+$-conductance(s) in NGFCs was not significantly inhibited by α-dendrotoxin (DTX; 100–200 nM), a specific blocker of Kv1 (Kv1.1, Kv1,2 and Kv1.6) channels (*Figure 4D*). A low concentration of 4-AP (low 4-AP; 100 μM), which also blocks Kv1 channels, caused only a modest yet statistically significant inhibition of subthreshold K$^+$-conductance(s) (*Figure 4D*). In contrast, millimolar concentrations of 4-AP (high 4-AP; 1–3 mM) produced a significant robust inhibition of the

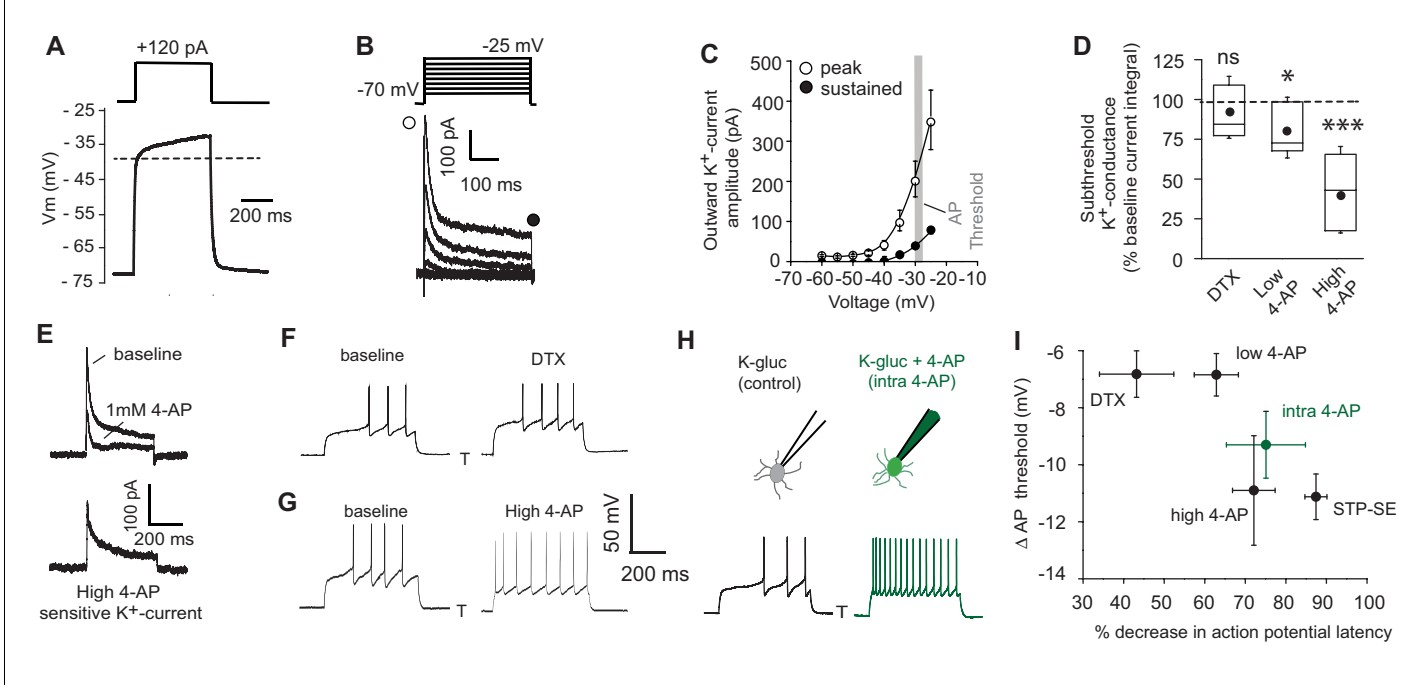

**Figure 4.** Pharmacological block of a subthreshold transient K$^+$-conductance mimics the AP latency and threshold modulation seen in STP-SE.
(A) Single example voltage trace to a depolarizing current injection (120 pA) in the presence of 1 µm TTX and 200 µm NiCl$_2$ to block voltage gated Na$^+$ and low-threshold activated Ca$^{2+}$ channels, respectively. (B) Representative voltage-gated outward K$^+$-currents activated in response to a series of predominantly subthreshold depolarizing voltage steps. Transient (open circle) and sustained (closed circle) components are labeled (C) Pooled data of the current-voltage curve of outward peak and sustained K$^+$-currents with gray shaded area depicting the mean action potential threshold measured in NGFCs (n = 6). (D) Box plot data depicting the pharmacological sensitivity of the subthreshold outward conductance to a subthreshold voltage step (−30 mV) by 100–200 nM DTX, 100 µm (low 4-AP) and 1–3 mM 4-AP (high 4-AP; n = 5, 7 and 11, respectively). p-values for DTX and high 4-AP datasets (Wilcoxin signed rank test) and paired t-test for low 4-AP were 0.44, 0.00098 and 0.02, respectively. (E) Single trace example of the pharmacological isolation of high 4-AP sensitive subthreshold outward K$^+$-current elicited in NGFCs characterized by a transient, inactivating time course. (F,G) Single trace examples showing the effect of DTX and high 4-AP on action potential output upon rheobase current step injections (120 pA). (H) Single trace examples comparing the effect of action potential output to a rheobase current injection performed with K$^+$-gluconate intracellular solution without (control; left trace) or with added 4-AP (intra-4-AP; right green trace; 3 mM). (I) Pooled data showing a comparison of the effects of DTX (n = 10), low 4-AP (n = 9), high 4-AP (n = 12), intra-4-AP (3 mM; n = 7) and STP-SE (n = 9) on action potential latency and threshold in relation to that seen with STP-SE (n = 9).

The online version of this article includes the following source data for figure 4:

**Source data 1.** Data plotted for *Figure 4*.

subthreshold K$^+$-conductance with this digitally isolated conductance exhibiting a distinct inactivating time profile reminiscent of A-type K$^+$-channels (I$_{KA}$; *Figure 4D,E*). We next examined the extent to which these pharmacological manipulations affected delay to spike and action potential threshold. Although DTX and low 4-AP reduced delay to spike and hyperpolarized action potential threshold, the extent to which this occurs was quite variable and not as marked to that seen following induction of STP-SE (*Figure 4F,I*). In contrast, high 4-AP and 4-AP delivered to NGFCs via the intracellular patch pipette (intra 4-AP; 3 mM) (*Jung and Hoffman, 2009*), modulated the delay to spike and hyperpolarized the AP threshold to an extent that more closely mimics the shift in both these parameters observed during STP-SE (*Figure 4G–I*).

Next, the ability of these pharmacological manipulations to increase intrinsic excitability in response to suprathreshold somatic current injections (1.5 × rheobase) was analyzed. In agreement with the propensity to modulate spike latency and AP threshold (*Figure 4I*), DTX and low 4-AP increased intrinsic excitability to a much lesser extent than high 4-AP or intra 4-AP (*Figure 5A-C*). Interestingly (in the presence of TTX) pharmacological inhibition of I$_{KA}$ with millimolar concentrations of 4-AP results in a transformation of the membrane voltage response characterized by the emergence of a depolarizing hump (*Figure 5D*) as would be predicted by the time-dependent profile of

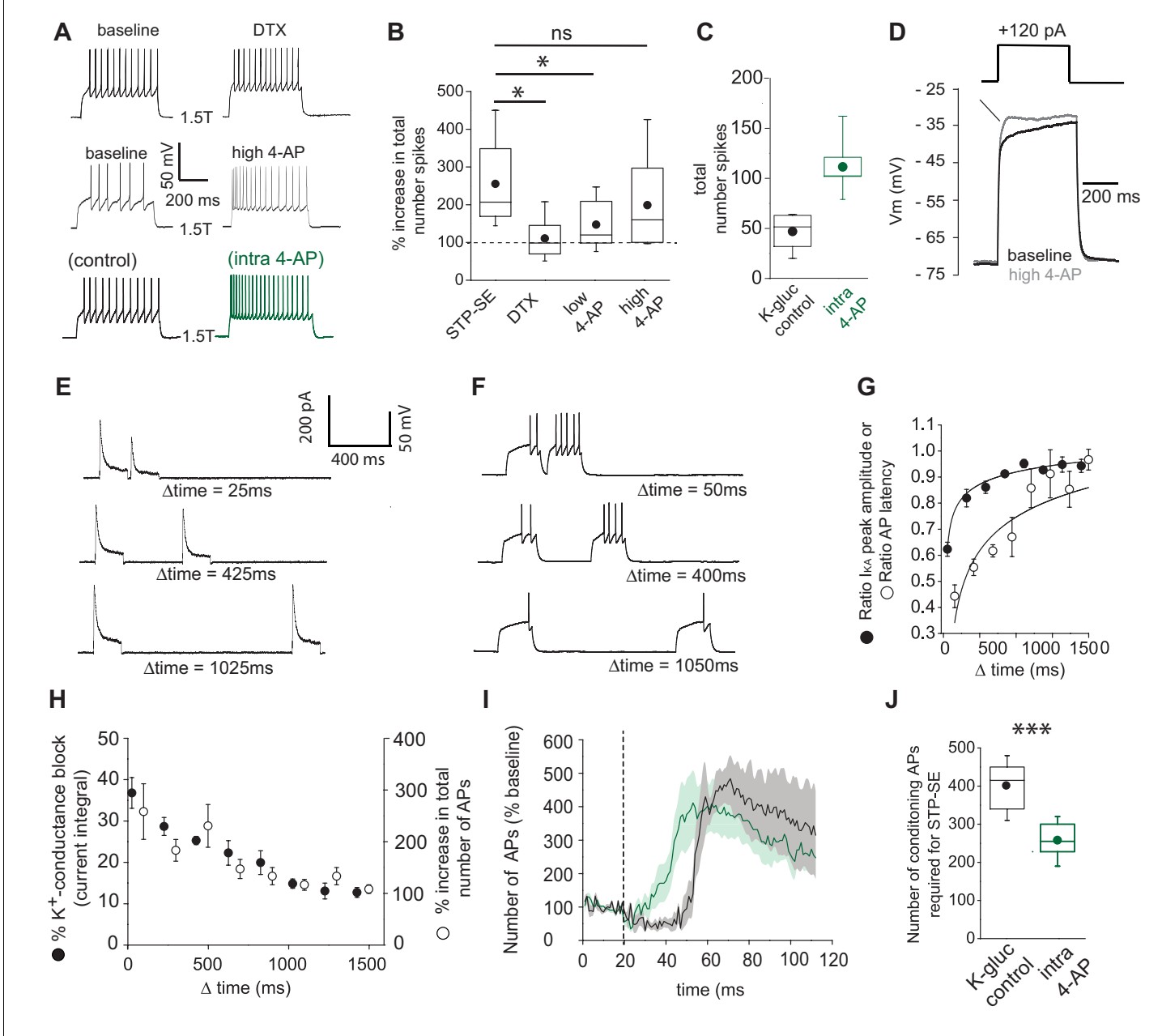

**Figure 5.** Functional availability of $I_{KA}$ modulates intrinsic excitability and induction requirements of STP-SE. (**A**) Single trace examples showing the effect of DTX (top traces), high 4-AP (middle traces) and with K+-gluconate intracellular solution without (control; bottom left trace) vs. 4-AP (intra-4-AP; bottom right green trace; 3 mM) on action potential output upon 1.5 x threshold current step injections (**B**) Pooled data showing the effects of STP-SE (n = 8), DTX (100–200 nM; n = 8), low 4-AP (100 μm; n = 9), high 4-AP (1 mM; n = 7) on intrinsic excitability measured the total number of action potentials to a series of depolarizing current injections. p-values from unpaired t-tests comparing increase in intrinsic excitability between STP-SE vs. DTX, STP-SE vs. low-4AP or STP-SE vs. high-4AP = 0.011, 0.045 and 0.38, respectively. (**C**) Pooled data comparing intrinsic excitability of NGFCs to series of depolarizing current injections with control intracellular K+-gluconate (n = 7; black trace) or K+-gluconate with added 4-AP (interleaved experiments; intra 4-AP 3 mM; n = 7; green trace). (**D**) Effect of high 4-AP (1 mM) on voltage response of NGFC to a subthreshold current injection step (120 pA). (**E**) Single trace examples illustrating recovery of subthreshold $I_{KA}$ current following successive depolarizing voltage steps of the same amplitude (typically 120 pA). (**F**) Corresponding action potential output of NGFCs to pairs of temporally separated threshold current injections (200 ms) shown previously in (**E**) to result in a range of $I_{KA}$ inactivation. (**G**) Pooled data showing comparison between the level of $I_{KA}$ inactivation induced measured as ratio of peak amplitude of $I_{KA}$ (n = 6) and ratio of latency (n = 6) of the first action potential between the two successive depolarizing voltage/current steps, respectively. (**H**) Pooled data showing relationship between the level of $I_{KA}$ inactivation induced (n = 6) and increase in action potential output (n = 6), the latter measured as percentage increase in number of action potentials of between the two successive depolarizing current steps. (**I**) Pooled time course of action potential output (measured as number of action potentials in response to the test rheobase current injection as

*Figure 5 continued on next page*

*Figure 5 continued*

percentage of baseline) during the entire STP-SE induction protocol in NGFCs with either control intracellular K⁺-gluconate (n = 7; black plot) or K⁺-gluconate with added 4-AP (interleaved experiments; intra 4-AP 3 mM; n = 7; green plot). Vertical dotted line indicates the commencement of induction with its termination occurring at varying times for each tested cell. (**J**) Box plot showing number of conditioning APs required for expression of STP-SE with control or intra-4AP K⁺-gluconate solution (interleaved experiments; n = 5 for both conditions; unpaired t-test p value = 0.00005).

The online version of this article includes the following source data for figure 5:

**Source data 1.** Data plotted for *Figure 5*.

$I_{KA}$ (*Figure 4B, E*). This increased envelope of depolarization, particularly at the onset of the depolarizing stimulus step, must contribute to the loss of delay to spike and increased excitability seen during STP-SE (*Figures 2A-E*, *3D,H,I*, *4G-I*).

We next took advantage of a characteristic biophysical property of $I_{KA}$ that manifests as a temporal delay in recovery of activation following inactivation to further investigate its role in NGFC excitablility. Thus, by delivering depolarizing subthreshold pulses in quick succession the extent of $I_{KA}$ influence in NGFCs could be experimentally 'titrated' (*Figure 5E,G*). In a separate series of experiments, we assessed the spike delay under conditions where the extent of $I_{KA}$ activation was manipulated in this manner (*Figure 5F,G*). We reveal that the time course of the recovery from inactivation of $I_{KA}$ is mirrored by that seen in the recovery of the delay to spike (*Figure 5G*). Furthermore, a close relationship exists between the extent of $I_{KA}$ inhibition in NGFCs (mediated by the differing magnitudes of recovery from inactivation) and corresponding increase in intrinsic excitability as measured by action potential output to the depolarizing test current injection (*Figure 5H*). Together, these pharmacological/biophysical analyses demonstrate that a transient, subthreshold K⁺-conductance(s) sensitive to high concentrations of 4-AP is a major contributor in dictating action potential latency, threshold and ultimately intrinsic excitability in NGFCs.

We next examined the extent to which pharmacological block of the $I_{KA}$ alters the ability of NGFCs to express STP-SE. For these experiments, $I_{KA}$ was blocked by introduction of 4-AP via the recording pipette as previously described (*Figure 4H*; *Figure 5A*) to prevent circuit wide perturbations that could confound the interpretation of results (*Deemyad et al., 2018*). Interleaved experiments (K-gluconate control vs. K-gluconate with intra 4-AP) clearly demonstrate that STP-SE could be induced more readily 4-AP containing internal solution as indicated by a leftward shift in the time course of potentiation that corresponds to a significant reduction in the number of required conditioning APs (*Figure 5I,J*). This gain of function provides further evidence for a reduction in 4-AP sensitive conductance(s) particularly those that resemble $I_{KA}$, as a contributing factor in STP-SE. However, due to our inability to fully occlude STP-SE we cannot rule out a role for additional modulation of other channels that are insensitive to intra-4-AP block.

## Functional $I_{KA}$ in mouse NGFCs is primarily mediated by Kv4 subunit containing channels

K⁺-channels comprised of distinct subunits can contribute to $I_{KA}$ in neural cell types (*Rudy, 1988*). Channels containing alpha subunits Kv1.1 (when in combination certain auxiliary subunits for example Kvβ1), Kv1.4, Kv3.3, Kv3.4 and Kv4 subunits can all mediate $I_{KA}$ and regulate neuronal excitability (*Carrasquillo et al., 2012*; *Rudy, 1988*). The complement of channels responsible for this conductance in NGFCs, however, has not been previously explored. Our previous pharmacological analyses to this point (*Figure 4*) suggest a relatively small contribution of Kv1.1 in increasing intrinsic excitability of NGFCs that itself is insufficient to fully recapitulate the changes observed in STP-SE (*Figure 5B*) thus implicating additional transient K⁺-channels. As a first pass to identify the molecular identity of $I_{KA}$ in NGFCs, we probed the publicly available Allen Brain Institute mouse dataset (*Tasic et al., 2018*) (https://portal.brain-map.org/atlases-and-data/rnaseq#Mouse_Cortex_and_Hip; see Materials and methods for details) containing single-neuron mRNA profiles of ~76,000 pan-glutamatergic and pan-GABAergic cortical and hippocampal neurons. Applying unsupervised graph-based clustering approaches using Seurat v3 (*Butler et al., 2018*; *Stuart et al., 2019*), we observed the cardinal neuronal classes (*Figure 6*; *Figure 6—figure supplement 1A,B*), including a clear segregation of MGE and CGE-derived GABAergic interneurons. Focusing attention on the 3626 neurons within the cluster corresponding to cortical and hippocampal CGE-NGFCs (*Figure 6A*; *Figure 6—figure supplement 1C*; see Materials and methods for details) these analyses reveal that

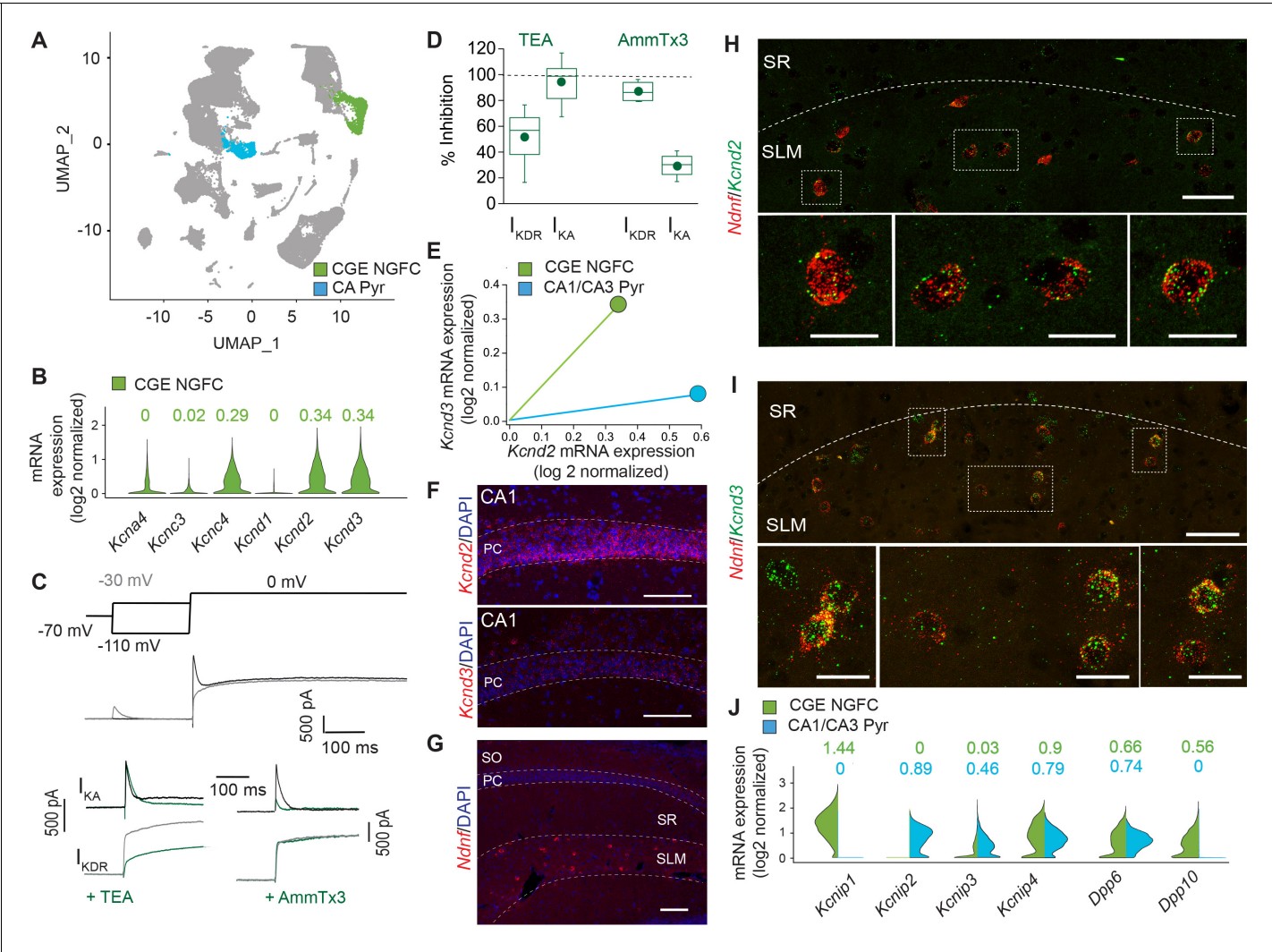

**Figure 6.** $I_{KA}$ in NGFCs is predominantly mediated by Kv4-subunit containing channels. (A) Uniform Manifold Approximation and Projection (UMAP) plot of RNAseq data publicly available from the Allen Brain Institute (see Materials and methods for details) highlighting clusters corresponding to CGE-NGFCs (green; 3626 cells) and hippocampal pyramidal cells (blue; 2530 cells). (B) Violin plots depicting comparison of mRNA levels in CGE-NGFCs that encode for subunits of $I_{KA}$ channels. Median values are depicted above each plot. (C) Top, middle; Voltage steps and corresponding single current traces illustrating the protocol employed to isolate $I_{KA}$ and $I_{KDR}$. Bottom; Single traces illustrating the effects of TEA (2–10 mM) and AmmTx3 (500 nM) on the isolated $I_{KA}$ and $I_{KDR}$. (D) Box plots depicting inhibition of $I_{KA}$ and $I_{KDR}$ by TEA (n = 8) and AmmTx3 (n = 4). (E) Scatterplot of median expression levels of *Kcnd2* vs. *Kcnd3* mRNA in CGE-NGFCs and hippocampal pyramidal cells. (F) Representative confocal images of *Kcnd2* and *Kcnd3* transcript expression in CA1 pyramidal cells revealed by RNAscope (see Materials and methods for details; scale bar = 100 µm) (G) Representative confocal image of the distribution of putative NGFCs (via expression of *Ndnf* mRNA transcripts) in CA1 SLM (scale bar = 100 µm). (H,I) Representative confocal images of mRNA expression of *Kcnd2* and *Kcnd3* in putative NGFCs (*Ndnf*-expressing) in CA1 SLM. (Scale bars = 50 µm for top panels; 20 µm for bottom panels) (J) Split-violin plots depicting comparison in the expression levels of mRNA transcripts encoding known auxiliary subunits of Kv4-containing channels (KChIPs and DPLPs) in CGE-NGFCs (green) vs. hippocampal pyramidal cells (blue).

The online version of this article includes the following source data and figure supplement(s) for figure 6:

**Source data 1.** Data plotted for *Figure 6*.
**Figure supplement 1.** Delineation of mouse cortico-hippocampal cell clusters obtained from publicly available Allen Brain Institute scRNAseq data.
**Figure supplement 1—source data 1.** Data plotted for *Figure 6—figure supplement 1*.
**Figure supplement 2.** Comparison of subthreshold Kv1 and Kv4 subunit gene expression between mouse and human NGFCs and other IN subtypes.

essentially no mRNA encoding for Kv1.4 (*Kcna4*), Kv3.3 (*Kcnc3*) and Kv4.1 (*Kcnd1*) subunits indicating their unlikely contribution to $I_{KA}$ in CGE-NGFCs (*Figure 6B*). In contrast, appreciable mRNA levels of *Kcnc4*, *Kcnd2* and *Kcnd3* mRNA that encode for Kv3.4, Kv4.2 and Kv4.3 subunits, respectively, were apparent (*Figure 6B*). We employed standard voltage-step protocols and digital subtraction to isolate sustained ($I_{KDR}$) and $I_{KA}$ currents in mouse hippocampal NGFCs (*Figure 6C*) to determine the relative contributions of these subunits in mediating $I_{KA}$ in NGFCs. As expected, TEA at the concentrations used (2–10 mM) reliably inhibits $I_{KDR}$ (*Figure 6C,D*). However, $I_{KA}$ in NGFCs is TEA insensitive indicating that Kv3.4 does not appreciably mediate NGFC $I_{KA}$ (*Rudy and McBain, 2001*) despite the presence of detectable levels of mRNA (*Figure 6C,D*). In contrast, $I_{KA}$ is largely abolished by the selective Kv4 blocker, AmmTx3 (*Maffie et al., 2013*) (% baseline $I_{KA}$ peak after 200 nM AmmTx3 = 29.6 ± 9.9%; n = 4; *Figure 6C,D*). Since the expression and functional role of Kv4-channels and their associated auxiliary subunits have been well characterized in principal cells, particularly those of the hippocampus (*Jerng and Pfaffinger, 2014*), we compared mRNA expression in these principal cells to that seen in CGE-NGFCs. In general, *Kcnd2*/Kv4.2 is predominantly expressed in PCs, particularly those in the CA1 region (*Figure 6E,F*; see Materials and methods for details), whereas *Kcnd3*/Kv4.3 is mainly restricted to INs (*Lien et al., 2002*; *Menegola et al., 2008*; *Rhodes et al., 2004*; *Serôdio and Rudy, 1998*). Here, the molecular profile analyses demonstrate that at least at the transcript level revealed by RNAseq analyses, NGFCs possess comparable levels of combined mRNA encoding *Kcnd2* and *Kcnd3* to that seen in PCs (*Figure 6E*). In agreement with the RNAseq and functional data, in situ hybridization using the RNAscope platform (see Materials and methods for details) revealed the presence of detectable *Kcnd2* and *Kcnd3* mRNA in putative SLM hippocampal NGFCs (identified by NDNF mRNA expression; *Figure 6G–I*). Finally, expression of mRNA for well-known Kv4 auxiliary subunits KChIPs and DPLPs in NGFCs (particularly Kcnip1, Kcnip4, *Dpp6* and *Dpp*10) albeit with a differential complement, is essentially similar if not higher to that observed in hippocampal pyramidal cells (*Figure 6J*). Interestingly, the presence of DPP6 or DPP10 subunits in the macromolecular complex of Kv4 channels is a requirement in conferring high AmmTx3 sensitivity (*Maffie et al., 2013*). Taken together the combined molecular, physiological and pharmacological approach demonstrate that the majority of mouse hippocampal CGE-NGFC $I_{KA}$ which strongly influences intrinsic excitability of this IN subtype, is most likely mediated via a combination of Kv4.2 and Kv4.3 containing channels.

## Modulation of dendritic integration by Kv4 channels and activity-dependent potentiation of E-S coupling in mouse NGFCs

Kv4 containing channels are located along the somatodendritic axis in numerous neuronal subtypes particularly studied in pyramidal cells where they influence membrane voltage responses to incoming inputs impinging on these domains (*Alfaro-Ruíz et al., 2019*; *Menegola et al., 2008*; *Trimmer, 2015*). Although the subcellular expression profile has not been resolved in NGFCs to date, we evaluated their potential functional role on the dendritic integration of EPSPs elicited in NGFCs. Electrical stimulation of SLM afferents elicited excitatory input onto NGFCs that displayed paired pulse facilitation in agreement with previous observations (*Chittajallu et al., 2017*; *Mercier et al., 2019*; *Price et al., 2005*; *Figure 7A*). Bath application of AmmTx3 had no significant effect on either the amplitude of EPSCs or paired pulse ratio, whereas a marked significant increase in the depolarization elicited by the same EPSCs when monitored as EPSPs in current clamp (*Figure 7A–D*). These data demonstrate that pharmacological modulation of Kv4 mediated $I_{KA}$ conductance in NGFCs alone, produces EPSP amplification facilitating the extent of temporal summation of excitatory afferent inputs onto NGFCs, independent to any changes in presynaptic release probability or postsynaptic glutamate receptor function. To this point in our study, we have assayed intrinsic excitability only in response to 'artificial' depolarizing current injection steps delivered to the soma. Based on our data thus far we would predict that if STP-SE occurs partly from a reduction of somatodendritic $I_{KA}$, an increase E-S coupling would be one physiological manifestation. Therefore, in a final set of experiments we probed the effect STP-SE on the integration of synaptic inputs onto NGFCs. EPSPs were evoked in NGFCs by electrical stimulation of SLM afferent fibers (*Figure 7E*). The NGFCs were then exposed to the induction protocol as previously described (*Figure 3A,B*) which was terminated immediately upon expression of STP-SE (*Figure 7F*). Electrical stimulation was recommenced at the same intensity as that used during baseline. In 10 out of 17 NGFCs tested, an increase in action potential output in response to synaptically evoked EPSPs was clearly evident (*Figure 7G,H*). This

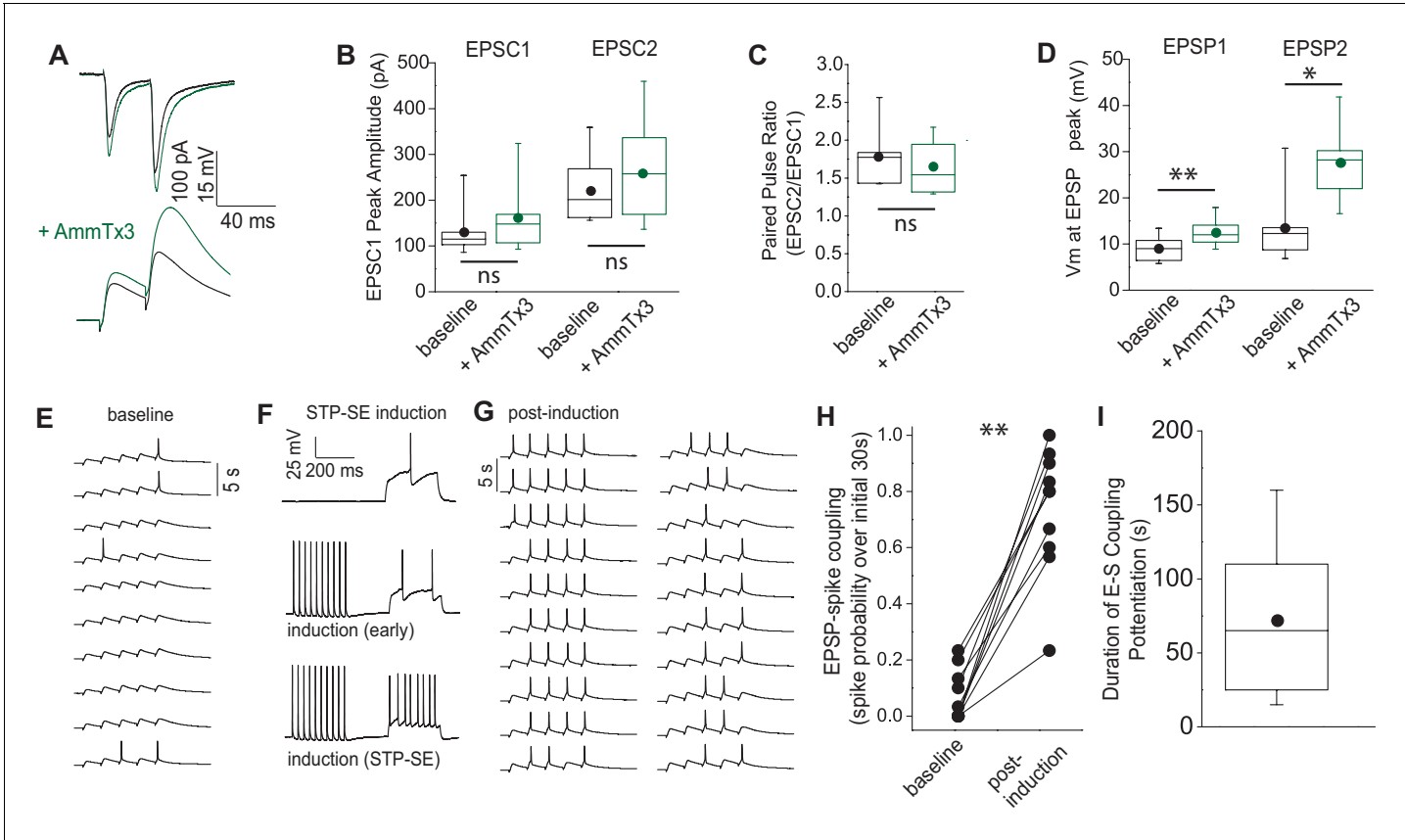

**Figure 7.** STP-SE manifests as an increase in afferent recruitment of NGFCs via enhanced E-S coupling. (**A**) Single traces showing the effect of AmmTx3 (500 nM) on EPSC (top traces) and corresponding EPSP (bottom trace) in a hippocampal NGFC in response to electrical stimulation of SLM afferent fibers (2 × 30 Hz). (**B**) Box plots of EPSC amplitudes (EPSC1 vs. EPSC2) under baseline conditions and after AmmTx3 treatment (n = 7; p values for EPSC1 and EPSC2 comparisons are 0.11 and 0.27, respectively). (**C**) Box plots of paired pulse ratio (EPSC2 peak amplitude/EPSC1 peak amplitude) under baseline conditions and after AmmTx3 treatment (n = 7; p value = 0.36). (**D**) Box plots of EPSP summation measured as absolute Vm at peak of EPSP1 and EPSP2 under baseline conditions and after AmmTx3 treatment (n = 7;=7; p values for EPSP1 and EPSP2 comparisons are 0.005 and 0.016, respectively)). (**E–G**) Single trace examples depicting a time course of E-S coupling (SLM afferent stimulation delivered at 5 × 30 Hz) during baseline conditions and after STP-SE induction. (**H**) Line plot depicting increased E-S coupling measured as spike probability during 10 sweeps of baseline vs. 10 sweeps immediately after STP-SE induction (n = 9; p value = 0.002). (**I**) Box plot illustrating duration of enhanced E-S coupling following STP-SE induction (n = 9).

The online version of this article includes the following source data and figure supplement(s) for figure 7:

**Source data 1.** Data plotted for *Figure 7*.
**Figure supplement 1.** E-S coupling potentiation not attributed to modulation of spontaneous EPSC parameters.
**Figure supplement 1—source data 1.** Data plotted for *Figure 7—figure supplement 1*.

potentiation of EPSP-spike (E-S) coupling possessed a similar mean duration to that seen in increased intrinsic excitability upon somatic depolarization following a single round of conditioning APs indicating similar underlying mechanisms (*Figure 7H* vs. *Figure 2G*; 63 ± 11 s vs. 77 ± 23 s for duration of STP-SE and E-S coupling potentiation, respectively; n = 10, 37; p=0.9; Mann-Whitney U - Test). In most recordings a biased current was imposed to ensure resting membrane voltage was equivalent between baseline and post STP-SE induction (Vm = −63.0 ± 0.7 mV vs. −62.7 ± 0.9 mV for baseline and post-induction, respectively; n = 10; p=0.52, Mann-Whitney U-Test). It must be noted that in these experiments, synaptic parameters such as probability of release and postsynaptic receptor function could potentially influence the extent of E-S coupling. Therefore, we also interrogated spontaneous EPSC parameters following induction of multiple BF episodes that have been shown to robustly condition the expression of STP-SE (*Figure 2G*). Interestingly, a small but significant decrease in sEPSC amplitude after the 4th BF episode (post-BF4) was noted, however, this would in fact *negatively* impact any potentiation noted and serve to minimize heightened E-S

coupling. In addition, no significant changes in sEPSC frequency or tau decay were observed (*Figure 7—figure supplement 1*). Together analyses of spontaneous EPSCs following induction of STP-SE confirm a modulation of intrinsic properties as the main underlying cause for the increased E-S coupling observed.

## Discussion

Varying mouse IN subtypes, including those in the NGFC family, possess the ability to demonstrate barrage or persistent firing (*Deemyad et al., 2018*; *Elgueta et al., 2015*; *Imbrosci et al., 2015*; *Krook-Magnuson et al., 2011*; *Rózsa et al., 2017*; *Sheffield et al., 2011*; *Sheffield et al., 2013*; *Suzuki et al., 2014*; *Wang et al., 2015*). BF is characterized by epochs of action potential output in the absence of depolarizing synaptic input thus circumventing canonical intraneuronal information transfer governed by neuronal polarity. However, induction of barrage firing typically requires protracted periods of action potential activity (*Sheffield et al., 2011*; *Suzuki et al., 2014*) (but see *Wang et al., 2015*; *Yoshida and Hasselmo, 2009*) questioning the exact physiological circumstances under which it manifests. Its operational role in normal brain function therefore remains unclear, although it has been postulated to constitute a mechanism for driving network oscillations (*Rózsa et al., 2017*), an important function of IN subclasses including NGFCs (*Capogna, 2011*; *Overstreet-Wadiche and McBain, 2015*; *Pelkey et al., 2017*). From a pathological standpoint, it has been proposed to constitute a homeostatic, dampening mechanism during prolonged epochs of circuit over excitation such as those that occur during epilepsy (*Elgueta et al., 2015*; *Suzuki et al., 2014*). Here we demonstrate the evolutionary retention of this IN subtype to enter this specific mode of intrinsic excitability and future work will be required to determine its exact role in human circuits. In addition to BF, we reveal the existence of a distinct and previously undescribed intrinsic plasticity mechanism expressed by both mouse and human NGFCs characterized by an increased efficacy of depolarizing input to elicit action potential output which we term STP-SE. We propose that, due to its relatively modest activity requirements, STP-SE possesses a greater propensity to operate under more physiological activity regimens Furthermore, it is envisaged that during the period of STP-SE, NGFCs can exhibit heightened inhibitory influence in response to afferent activity patterns driven by relevant stimuli and ongoing behavior.

In agreement with a previous study (*Poorthuis et al., 2018*), our observations demonstrate a heightened basal human NGFC excitability when compared to rodent (*Figure 2C*; open black vs.s open orange circles). It is unclear whether this reflects a true divergence in intrinsic properties between mouse and human NGFCs. Although we procured human tissue from brain regions distal to the epilepsy focal point that was deemed non-pathological (see Materials and methods for details), we cannot discount possible modulation of neuronal function precipitated by patient treatment regimens. However, NGFCs in control non-human primate brain also demonstrate a similar heightened basal excitability (*Povysheva et al., 2007*) suggesting that this indeed reflects an evolutionary distinction. Nevertheless, regardless of differences in basal excitability and their underlying reasons, our data clearly demonstrate both mouse and human NGFCs to undergo essentially similar intrinsic plasticity. It is highly unlikely that either disease progression and/or drug treatment in humans would result in species mirroring in the propensity to express such an activity-dependent modulation described. Therefore, although the ability of non-human primate NGFCs to undergo STP-SE (and BF) is unknown, we are confident that the intrinsic plasticity demonstrated by human NGFCs are not a consequence of disease state and/or drug regimens.

In the current study, we show that mouse NGFCs express Kv4 and auxiliary subunit transcripts that translates to a functional $I_{KA}$ conductance. Although the functional ramifications of Kv4 channel and auxiliary subunits have been mainly investigated in pyramidal cells, their expression and role in modulating subthreshold membrane dynamics, firing properties and dendritic excitability of IN subtypes have been described (*Bourdeau et al., 2011*; *Bourdeau et al., 2007*; *Goldberg et al., 2003*; *Lien et al., 2002*; *Oláh et al., 2020*; *Pelkey et al., 2017*; *Rhodes et al., 2004*; *Sun, 2009*; *Wang et al., 2004*; *Williams and Hablitz, 2015*). We extend these observations to NGFCs and demonstrate $I_{KA}$ can effectively influence delay to spike and action potential threshold resulting in increased intrinsic excitability like that seen in STP-SE ultimately resulting in enhanced dendritic integration. This coupled with the fact that STP-SE is more readily induced under conditions where $I_{KA}$ is inhibited points to a potential modulation of this conductance in the expression of intrinsic plasticity

described in this study. However, care must be taken in interpretation of these data and further experimentation will be required to directly implicate the modulation of $I_{KA}$ mediated by Kv4-containing channels as the sole contributor underlying the expression of STP-SE in NGFCs. As such, we cannot presently exclude additional contributions of other channels expressed along the axodendritic axes (e.g. KCNQ, HCN, ether-a-go-go, SK/BK channels), a complex interplay of which, could contribute to produce the enhanced excitability and E-S coupling described here.

Extrapolating mechanistic studies performed in mouse to human especially in light of recent observations highlighting divergence of molecular properties between equivalent IN subtypes across evolution (*Hodge et al., 2019*; *Krienen et al., 2019*) also requires caution. For example, examination of available RNAseq data from mouse and human via available transcriptomic explorers (*Hodge et al., 2019*; *Krienen et al., 2019*) (http://celltypes.brain-map.org/rnaseq/mouse_ctx-hip_smart-seq.; http://celltypes.brain-map.org/rnaseq/human_ctx_smart-seq. ; http://interneuron.mccarrolllab.org/) reveals essentially a complete absence of *Kcna1* mRNA levels in human IN populations (*Figure 6—figure supplement 2*). Indeed, a number of studies have highlighted an important role of Kv1.1 for mouse IN excitability (*Campanac et al., 2013*; *Debanne et al., 2019*; *Gainey et al., 2018*; *Li et al., 2012*). Thus, these cross-species transcriptomic profiling data questions to what extent its functional modulation is relevant for shaping IN excitability within humans. As previously discussed, Kv1.1 plays a minimal role with a relatively larger and measurable impact of Kv4-channel mediated $I_{KA}$ in dictating mouse NGFC excitability. Interestingly, gene expression of *Kcnd2*, *Kcnd3* (and associated auxiliary subunit mRNA; https://portal.brain-map.org/; http://interneuron.mccarrolllab.org/) are conserved across evolution (*Figure 6—figure supplement 2*). These gene expression studies (*Hodge et al., 2019*; *Krienen et al., 2019*) indicate a maintained presence of Kv4 channels at the transcript level at least. However, a direct functional role for these subthreshold conductances in dictating human NGFC excitability akin to that in mouse remains to be experimentally ascertained.

At the cellular level, dendritic inhibition exerts a strong influence on excitability and electrogenesis in distal pyramidal cell subdomains (*Kepecs and Fishell, 2014*; *Pelkey et al., 2017*). At the network level this translates into changes in efficiency of pyramidal cell recruitment, gating of information transfer within neural circuits, sculpting the propensity for induction/expression of synaptic plasticity and coordinating pyramidal cell ensembles to generate specific oscillatory events that are known to underly cognitive processing. Given these critical network roles, it is not surprising that various behaviors are encoded through the activity and modulation of dendritic targeting INs (*Cummings and Clem, 2020*; *Lovett-Barron et al., 2014*; *Abs et al., 2018*). However, although constituting one of the major populations of INs providing dendritic inhibition (*Bezaire and Soltesz, 2013*) limited information is available regarding the in vivo role of the NGFC subfamily. Recently, a correlation between the recruitment of inhibition mediated by NDNF-expressing cells (a subpopulation corresponding to NGFCs [*Abs et al., 2018*; *Tasic et al., 2016*] and conditioned learning in mice has provided the first description of a link between NGFC function and behavior (*Abs et al., 2018*). The control of dendritic excitability, including that mediated by Kv4 channels, alters the propensity for LTP induction/expression of excitatory afferent input onto pyramidal cells (*Kim and Hoffman, 2008*). Interestingly, mouse NGFCs also exhibit Hebbian long-term plasticity (LTP) triggered by $Ca^{2+}$-influx via NMDA receptor and/or VGCC activation (*Mercier et al., 2019*). NGFCs located in superficial regions of cortex and hippocampus are impinged upon via multiple glutamatergic (e.g. thalamic and cortical) and neuromodulatory afferent populations (*Abs et al., 2018*; *Chittajallu et al., 2017*; *Lee et al., 2010*; *Mercier et al., 2019*) each conveying distinct information. Thus, it is plausible that increased intrinsic excitability occurring in response to the activity of a given specific afferent input can render a second distinct input more efficacious in NGFC recruitment for short- or long-term periods. The resultant modulation of dendritic inhibition driven by afferent inputs onto NGFCs can potentially gate synaptic plasticity of excitatory transmission onto pyramidal cells. Together, these outlined metaplastic routes bestow the microcircuit with a complex array of cellular mechanisms relevant for top-down processing and encoding of contextual information underlying specific learning and memory tasks (*Abs et al., 2018*).

Here, we focused on mouse and human NGFCs that reside in the most superficial regions of cortex and hippocampus. These NGFCs are largely derived from an embryonic progenitor pool originating in the caudal ganglionic eminence (CGE) (*Overstreet-Wadiche and McBain, 2015*) *Pelkey et al., 2017* #55}. Another distinct group of NGFCs (along with the closely related Ivy

cells *Overstreet-Wadiche and McBain, 2015*; *Pelkey et al., 2017* have been identified particularly in mouse hippocampus and can be delineated based on a distinct developmental origin in the medial ganglionic eminence (MGE). Intriguingly, recent cross-species transcriptomic analyses have revealed a remarkable expansion (in terms of number relative to the other IN subtypes) of MGE-derived NGFCs that appear in deeper layers of human sensory and associational cortices when compared to equivalent brain regions in mouse (*Krienen et al., 2019*). Whether this expanded population of MGE-derived cortical NGFC population in humans undergo similar intrinsic property modulation to their CGE counterparts described here remains to be determined. If so, this would markedly increase the available substrate, further expanding the influence of the described plasticity in cortical circuits of humans. A full description of the physiological properties and detailed mechanistic studies regarding human IN function are hindered due to an inability to directly label them in live human tissue, particularly in studies where a specific subtype is of primary interest. Recently, viral strategies have been successfully developed permitting the fluorescent tagging of IN subtypes in non-human primate and human tissue (*Dimidschstein et al., 2016*; *Mehta et al., 2019*; *Mich et al., 2020*; *Vormstein-Schneider et al., 2020*).

In summary, although our data point to a role for modulation in $I_{KA}$ mediated by Kv4 channels, further elucidation of the cellular mechanisms underlying the described intrinsic plasticity expressed by NGFCs and to what extent these are conserved across evolution will require further work as discussed. However, regardless of the exact mechanistic underpinnings, we clearly demonstrate that both mouse and human NGFCs possess a remarkably similar propensity to modulate their intrinsic properties in the face of ongoing activity. The evolutionary retention of these mechanisms reveals circuit motifs that represent critical facets of human brain function. From a research perspective this conservation highlights the mouse model as an applicable tool (*Abs et al., 2018*) to predict the potential human behavioral correlates dependent on the tuning of inhibition mediated by this subtype of specialized neuron. Refinement of the above viral strategies to incorporate the capacity for genetic manipulation (*Mehta et al., 2019*; *Vormstein-Schneider et al., 2020*) will greatly facilitate further studies aimed at elucidating the functional role of INs, including those mediated by the diverse members (i.e. with respect to embryonic origin) of the NGFC subfamily, embedded in human neural circuitry.

## Materials and methods

### Tissue procurement

Adult (p35-p60) male and female *Gad2*-EGFP (Tg(Gad2-EGFP)DJ31Gsat/Mmucd; GENSAT; Cat. No. 011849-UCD) or *Htr3a*-EGFP (Tg(*Htr3a*-EGFP)DH30Gsat/Mmnc; GENSAT; Cat. No. 000273-UNC) mice were used and all experiments were conducted in accordance with animal protocols approved by the National Institutes of Health (ASP# 17–045). For human data, we collected surgical specimens from ten participants (seven male; age range 22–69 years) with drug-resistant epilepsy who underwent an initial surgical procedure in which intracranial recording electrodes were implanted subdurally on the cortical surface as well as within the brain parenchyma for seizure monitoring followed by a second surgical procedure in which a resection was performed. In each case, the clinical team determined the initial placement of the contacts to localize the epileptogenic zone. Based on the intracranial recordings during the monitoring period, the clinical team then determined which electrode contacts lay over brain regions exhibiting ictal or inter-ictal activity and which contacts were electrographically normal. The surgical resections during the second surgery were designed to remove the areas of seizure onset, but in every case, also involved removal of some brain tissue determined to be electrographically normal. In eight participants, we collected electrographically normal brain specimens from the lateral temporal lobe cortex which was removed as part of a standard anterior temporal lobectomy for resection of the hippocampus. In the two remaining participants, we obtained parietal and frontal lobe brain specimens that were immediately adjacent to the epileptogenic zone but that were also determined to be electrographically normal. The NINDS Institutional Review Board (IRB) approved the research protocol (ClinicalTrials.gov Identifier NCT01273129), and informed consent for the experimental use of resected tissue was obtained from each participant and their guardians.

## Electrophysiology

Both mouse and human brain tissue were sectioned at 300 μm using a Leica Vibratome 1200s in ice-cold sucrose-based cutting solution of the following composition (mM): 90 sucrose, 80 NaCl, 3.5 KCl, 24 NaHCO$_3$, 1.25 NaH$_2$PO$_4$, 4.5 MgCl$_2$, 0.5 CaCl$_2$, and 10 glucose, saturated with 95% O$_2$ and 5% CO$_2$. Slices were allowed to equilibrate in warm (31–33°C) sucrose cutting solution for 20–30 min after which the solution was allowed to return to room temperature where they were maintained until electrophysiological analyses. Electrophysiological recordings were performed on slices perfused with extracellular solution of the following composition (mM): 130 NaCl, 24 NaHCO$_3$, 1.25 NaH$_2$PO$_4$, 3.5 KCl, 1.5 MgCl$_2$, 2.5 CaCl$_2$ and 10 glucose saturated with 95% O$_2$ and 5% CO$_2$. In certain experiments (see results section for details) the extracellular solution was supplemented with one or a combination of the following drugs; 50 μM picrotoxin (Millipore Sigma; Cat. No. 80410), 2–5 μM CGP556845A (Abcam; Cat. No. ab120337), 5–10 μM bicuculline methobromide (Abcam; Cat. No. ab120109), 100 μM or 1–3 mM 4-aminopyridine (Millipore Sigma; Cat. No. A78403), 100–200 nM α-DTX (Alomone Labs; Cat No.D-350), 200 or 500 nM AmmTx3 (Alomone Labs; Cat No.STA-305). Whole cell voltage and current-clamp recordings were performed at 32–34°C with electrodes (3–5mΩ) pulled from borosilicate glass (World Precision Instruments, Sarasota, FL) filled with intracellular solution of the following composition (in mM); 150 K–gluconate, 3 MgCl$_2$, 0.5 EGTA, 2 MgATP, 0.3 Na$_2$GTP, 10 HEPES and 3 mg/ml biocytin; pH was adjusted to 7.3 with KOH and osmolarity adjusted to 280–300 mOsm. All electrophysiological recordings were performed using Multiclamp 700B amplifier (Molecular Devices, Sunnyvale, CA, USA). Signals were filtered at 4–10 kHz and digitized at 10–20 kHz (Digidata 1322A and pClamp 10.2 Software; Molecular Devices, Sunnyvale, CA, USA). For analyses of K$^+$-channel currents in NGFCs, voltage clamp recordings were performed in the presence of 0.5 μm TTX and 200 μM NiCl$_2$ with voltage steps delivered as indicated in the manuscript. Digital leak subtraction was performed online (P/4) to remove capacitance transient and leak currents and series resistance compensation was not performed. Glutamate receptor-mediated synaptic responses (EPSCs and EPSPs) were evoked with A360 constant-current stimulus isolator (World Precision Instruments, Sarasota, FL, USA) and with stimulation electrodes pulled from borosilicate glass filled with aCSF placed in SLM in the presence 50 μM picrotoxin, 2–5 μM CGP556845A and 5–10 μM bicuculline to block GABA receptor-mediated responses. In experiments assessing the effect of AmmTX3 on EPSP integrations 5 mM QX-314Cl$^-$ was included in the intracellular solution to prevent activation of voltage-gated sodium channels.

For neuron morphology recoveries after electrophysiological recordings, slices were post-fixed in 4% paraformaldehyde in 0.1M phosphate buffer at least overnight followed by fluorescent conjugation to biocytin and confocal images attained using a Zeiss 780 microscope (NICHD Microscopy and Imaging Core) and reconstructions drawn using Neurolucida software (MBF Bioscience, VT).

## Fluorescence in situ hybridization employing the RNAscope platform

In this study, two mice aged p35 and p40 were anesthetized with isoflurane and intracardially perfused with 0.1M phosphate buffer, Ph7.4, followed by 4% paraformaldehyde/0.1M phosphate buffer. Brains were removed and post-fixed in 4% paraformaldehyde/0.1M phosphate buffer overnight, then embedded in Tissue-Tek O.C.T. compound {Sakura Finetek, USA Cat. No 4583} and frozen. 10–14 μM horizontal sections containing mid-ventral hippocampus were cut using a Leica CM3050-S cryostat, mounted on microscope slides and stored at −80°C until use. Fluorescent in situ hybridization was performed according to the RNAscope (Advanced Cell Diagnostics USA, Newark, CA) protocol using the Multiplex Fluorescent Reagent Kit v1 (Cat. No 320851). Probes employed were Mm-Ndnf (Cat No. 447471-C2), Mm-Kcnd2 (Cat No. 452581) and Mm-Kcnd3 (Cat No. 573641). Sections were cover-slipped and confocal images taken using Zeiss 780 microscope (NICHD Microscopy and Imaging Core). Z-stack images were acquired using a 20x/0.8 Plan Apochromat or 40x/1.3 Plan Neofluar objective and were rendered as maximum intensity injections.

## Analyses of single cell RNAseq (scRNAseq) transcriptomic data available from the Allen Brain Institute

To analyze the Allen Institute mouse dataset of the single-cell transcriptomes of ~76,000 cells from >20 areas of mouse cortex and hippocampus, we downloaded the transcript.tome/HDF5 file (https://portal.brain-map.org/atlases-and-data/rnaseq) and subsequently converted into Seurat v3-

compatible format based on the instructions provided in the Allen Institute Portal (https://portal.brain-map.org/atlases-and-data/rnaseq/protocols-mouse-cortex-and-hippocampus) and custom scripts in R package. Basic processing and visualization of the scRNA-seq data were performed with the Seurat v3 in R (*Butler et al., 2018*; *Stuart et al., 2019*). Briefly, data were preprocessed by removing cells with less than 200 detectable genes and with reads mapping to mitochondrial genes comprising more than 0.05% of total reads. Gene expression counts were then log normalized and the resulting expression matrix was scaled to 10,000 transcripts per cell. Next, the expression values of each gene across all the cells (z-score transformation) were utilized for dimensionality reduction. These were performed using the NormalizeData() and ScaleData() functions in Seurat with default parameters. Next, we identified the top 2000 genes that exhibited high variability across cells and therefore represent unique features of the cell clusters, for downstream analysis using the FindVariableGenes() function, by applying selection.method = 'vst' feature. Next, principal component analysis (PCA) was performed, and the top 20 principal components were stored. We clustered the cells using the FindClusters() with a clustering resolution set to 0.6. This approach generated a total of 38 clusters. Using the FindAllMarkers() function at cutoff of genes expressed in 25% of cells, at a logfc. threshold = 0.25, we identified the genes that are expressed in each cluster. For each cell type we used the well-established marker genes that uniquely established the cell identity, as previously described in the literature. Notably, *Snap25* positive clusters were annotated as neurons, and the *Snap25* negative clusters were annotated as non-neurons; among the neurons, *Slc17a7* was indicative of glutamatergic and *Gad1* expression indicative of GABAergic neurons (*Figure 6—figure supplement 1A*). More specifically, the *Gad1* clusters expressing a combination of *Lhx6, Pvalb, Sst, Hapln1* are indicative of MGE-derived GABAergic interneurons, and a combination of *Lhx6*-negative, *Prox1, Vip, Calb2, Cnr1, Hapln1, Ndnf* are indicative of CGE-derived GABAergic interneurons (*Figure 6—figure supplement 1B*). Relevant for the present study, specific-markers include, but are not limited to: *Gad1, Prox1, Hapln1, Lamp5, Ndnf,* but *Lhx6* negative, for the CGE-NGFCs (*Mayer et al., 2018*; *Tasic et al., 2018*); *Slc17a7, Wfs1, Fibcd1, Crym* for the hippocampal pyramidal neurons (*Cembrowski et al., 2016a*; *Cembrowski et al., 2016b*). The expression of *Kcn* genes and auxiliary subunits were obtained for the above clusters, and the Violin/Split-violin plots were plotted via custom scripts in the R package using ggplot. All scripts employed are available at GitHub https://github.com/mahadevanv/Chittajallu-et-al-Allen-Mouse-CGE-NGFCs (*Chittajallu, 2020*; copy archived at https://github.com/elifesciences-publications/Chittajallu-et-al-Allen-Mouse-CGE-NGFCs).

## Statistics

Each dataset that underwent statistical analyses was first tested for normal distribution using the Shapiro-Wilk test. For data that were deemed normally distributed parametric paired or unpaired t-tests were employed, otherwise Mann-Whitney U-tests or Wilcoxin-sign Rank tests were used. Due to the low n numbers in some human datasets (*Figure 2C*, n = 3; *Figure 3H,I*, n = 4), Shapiro-Wilk or non-parametric tests could not be performed and in these cases we defaulted to parametric Students T-test. In the figure panels statistical significance is denoted as follows; *p<0.05, **p<0.01, ***p<0.001, ns not significant p>0.05. The type of statistical test and exact p-values are indicated in the figure legends where appropriate. n values throughout the manuscript correspond to the number of cells tested and for each set of experiments and are from a minimum of 3 mice and two human subjects. Throughout the manuscript figures, data in black/grey and orange were from mouse and human, respectively. In box plots, the data are represented as the median and mean (line and symbol, respectively), the 25th and 75th percentile (bottom and top of the box, respectively), 10th and 90th percentiles (bottom and top of whiskers, respectively). In scatter and line plots all data are expressed as the mean ± s.e.m.

## Acknowledgements

We thank Apratim Mitra and Ryan Dale (NICHD Bioinformatics and Scientific Programming Core) and Christopher T Rhodes (Dr. Timothy Petros' Laboratory, Unit on Cellular and Molecular Development, NICHD, NIH) for assistance with analyzing the Allen Brain scRNAseq dataset. These analyses utilized the computational resources of the NIH PC Biowulf cluster (http://hpc.nih.gov). We are grateful to Lynne Holtzclaw (NICHD Microscopy and Imaging Core) for invaluable help with tissue

preparation, RNAscope and imaging of fluorescent in situ hybridization . We also thank Cameron Paranzino for performing Neurolucida reconstructions of biocytin filled neurons.

## Additional information

### Funding

| Funder | Grant reference number | Author |
| --- | --- | --- |
| NINDS Intramural Research Program | Z01NS003144 | Kareem A Zaghloul |
| NICHD Intramural Research Program | ZIAHD001205 | Chris J McBain |

The funders had no role in study design, data collection and interpretation, or the decision to submit the work for publication.

### Author contributions

Ramesh Chittajallu, Conceptualization, Data curation, Formal analysis, Supervision, Investigation, Visualization, Methodology, Writing - original draft, Project administration, Writing - review and editing; Kurt Auville, Formal analysis, Investigation; Vivek Mahadevan, Software, Formal analysis, Investigation, Visualization, Writing - original draft, Writing - review and editing; Mandy Lai, Investigation; Steven Hunt, Daniela Calvigioni, Investigation, Methodology; Kenneth A Pelkey, Conceptualization, Supervision, Writing - review and editing; Kareem A Zaghloul, Resources, Funding acquisition, Neurosurgical resection of human tissue samples from patients with intractabl epilepsy; Chris J McBain, Conceptualization, Supervision, Funding acquisition, Project administration, Writing - review and editing

### Author ORCIDs

Ramesh Chittajallu (iD) https://orcid.org/0000-0002-9794-0052
Vivek Mahadevan (iD) http://orcid.org/0000-0002-0805-827X
Kenneth A Pelkey (iD) http://orcid.org/0000-0002-9731-1336
Kareem A Zaghloul (iD) http://orcid.org/0000-0001-8575-3578
Chris J McBain (iD) https://orcid.org/0000-0002-5909-0157

### Ethics

Human subjects: The NINDS Institutional Review Board (IRB) approved the research protocol (ClinicalTrials.gov Identifier NCT01273129), and informed consent for the experimental use of resected tissue was obtained from each participant and their guardians.
Animal experimentation: All mice were handled in accordance with animal protocols approved by the National Institutes of Health (ASP# 17-045).

### Decision letter and Author response

Decision letter https://doi.org/10.7554/eLife.57571.sa1
Author response https://doi.org/10.7554/eLife.57571.sa2

## Additional files

### Supplementary files

• Transparent reporting form

### Data availability

All data generated or analysed during this study are included in the manuscript and supporting source data files.

The following previously published datasets were used:

| Author(s) | Year | Dataset title | Dataset URL | Database and Identifier |
|---|---|---|---|---|
| Hodge RD, Bakken TE, Miller JA, Smith KA, Barkan ER, Graybuck LT, Close JL, Long B, Johansen N, Penn O, Yao Z, Eggermont J, Höllt T, Levi BP, Shehata SI, Aevermann B, Beller A, Bertagnolli D, Brouner K, Casper T, Cobbs C, Dalley R, Dee N, Ding S, Ellenbogen RG, Fong O, Garren E, Goldy J, Gwinn RP, Hirschstein D, Keene CD, Keshk M, Ko AL, Lathia K, Mahfouz A, Maltze Z, McGraw M, Nguyen TN, NyhusJ, Ojemann JG, Oldre A, Parry S, Reynolds S, Rimorin C, Shapovalova NV, Somasundaram S, Szafer A, Thomsen ER, Tieu M, Quon G, Scheuermann RH, Yuste R, Sunkin SM, Lelieveldt B, Feng D, Ng L, Bernard A, Hawrylycz M, Phillips JW, Tasic B, Zeng H, Jones AR, Koch C, Lein S | 2019 | Cell Diversity in the Mouse Cortex and Hippocampus | https://transcriptomic-viewer-downloads.s3-us-west-2.amazonaws.com/mouse/transcriptome.zip | Allen Brain Map, mouse/transcriptome |
| Hodge RD, Bakken TE, Miller JA, Smith KA, Barkan ER, Graybuck LT, Close JL, Long B, Johansen N, Penn O, Yao Z, Eggermont J, Höllt T, Levi BP, Shehata SI, Aevermann B, Beller A, Bertagnolli D, Brouner K, Casper T, Cobbs C, Dalley R, Dee N, Ding S, Ellenbogen RG, Fong O, Garren E, Goldy J, Gwinn RP, Hirschstein D, Keene CD, Keshk M, Ko AL, Lathia K, Mahfouz A, Maltze Z, McGraw M, Nguyen TN, NyhusJ, Ojemann JG, Oldre A, Parry S, Reynolds S, Rimorin C, Shapovalova NV, Somasundaram S, Szafer A, Thomsen ER, Tieu M, Quon | 2019 | Cell Diversity in the Human cortex | https://transcriptomic-viewer-downloads.s3-us-west-2.amazonaws.com/human/transcriptome.zip | Allen Brain Map, human/transcriptome |

G, Scheuermann RH, Yuste R, Sunkin SM, Lelieveldt B, Feng D, Ng L, Bernard A, Hawrylycz M, Phillips JW, Tasic B, Zeng H, Jones AR, Koch C, Lein S

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
