## [Decision Letter]

**Acceptance summary:**

Retroaxonal barrage firing contradicts typical organizing principles of neuronal networks, as sustained activity generated self-autonomously in axons bypasses the classical dendrite-to-axon information route to allow sustained action potential activity in the absence of synaptic activity. Here the authors provide new insight into the consequences of barrage firing, showing that it generates plasticity of intrinsic excitability of both mouse and human neurogliaform interneurons.

**Decision letter after peer review:**

Thank you for submitting your article "Activity-dependent tuning of intrinsic excitability in mouse and human neurogliaform cells" for consideration by *eLife*. Your article has been reviewed by three peer reviewers, including Linda Overstreet-Wadiche as the Reviewing Editor and Reviewer #1, and the evaluation has been overseen by Gary Westbrook as the Senior Editor. The following individuals involved in review of your submission have agreed to reveal their identity: János Szabadics (Reviewer #2); Johannes J Letzkus (Reviewer #3). The reviewers have discussed the reviews with one another and the Reviewing Editor has drafted this decision to help you prepare a revised submission.

We would like to draw your attention to changes in our revision policy that we have made in response to COVID-19 (https://elifesciences.org/articles/57162). Specifically, when editors judge that a submitted work as a whole belongs in *eLife* but that some conclusions require a modest amount of additional new data, as they do with your paper, we are asking that the manuscript be revised to either limit claims to those supported by data in hand, or to explicitly state that the relevant conclusions require additional supporting data. Our expectation is that the authors will eventually carry out the additional experiments and report on how they affect the relevant conclusions either in a preprint on bioRxiv or medRxiv, or if appropriate, as a Research Advance in *eLife*, either of which would be linked to the original paper.

Summary:

The authors of this manuscript address a new form of plasticity of intrinsic excitability in mouse and human neurogliaform cells (NGFCs). Using acute slice recordings, they demonstrate that barrage firing, previously found in mouse NGFCs, is also observed in human NGFCs and that it induces short-term potentiation of intrinsic excitability (STP-SE) in both species. The authors go on to investigate the complement of voltage-gated potassium channels expressed by mouse NGFCs that dictate excitability and action potential latency, showing that blocking IA enhances excitability and facilitates STP-SE induction. They analyze published data sets and their own in situ hybridizations to identify Kv4-containing channels as the likely substrate of IA in NGFCs. In a final set of experiments, they demonstrate that IA suppresses, and STP-SE induction enhances, E-S coupling for synaptic inputs. The reviewers agreed that this work provides new understanding of NGFCs and the side-by-side comparison of mouse and human data confirms that the described phenomenon is robust through mammalian evolution. The experiments are innovative and performed to high standards, and reported in a clear and transparent fashion. However, there are a number of issues, both essential and minor, that we ask that you address in the revision.

Essential revisions:

1) The AP peak changed in parallel with the threshold on Figure 3F-G. There is no obvious solution for these two changes because a shift in the threshold implies more efficient sodium channel activation, whereas decrease in the peak of AP implies less efficient sodium channel function. It is important to discuss this paradox, because some technical issues (e.g. offset change) can cause parallel shifts of the two parameters. At least the authors should demonstrate that there are no major changes in the recording conditions by providing data about Rin and Rs parameters. They could also potentially show that the observed effects are specific to AP parameters and that the voltage dependence of independent mechanisms are not changed by the induction protocol, such as the reversal potentials of potassium and sodium (or the more easily measured AMPA and GABA currents).

2) It appears in both species that not only the threshold has changed after STP-SE induction, but axonal and somatic components of the AP become more separated (Figure 3FG). This change may indicate a shift in the electrotonic distance of the AP initiation zone from the soma. As the position of AP initiation is crucial for the general excitability, this parameter should be quantified on the available recordings by using the 2nd derivative of the APs. If there is indeed a difference on the whole data set (not only on these two examples), the authors should provide additional discussion how this might contribute to STP-SE.

3) The data from mice was obtained from CGE-originated NGFCs as a consequence of using of specific mouse lines. Thus, MGE-derived NGFCs were mostly excluded from the mouse analysis whereas the origin of human NGFCs could not be determined. The authors also discussed that human NGFCs may show larger diversity than NGFCs in rodents but it is not clear how this larger variability contribute to some of the differences between human and mice NGFCs that were observed in this study. Therefore, it would be important to determine the homogeneity of human NGFCs and/or compare with that of mouse NGFCs. This question is particularly relevant when hippocampal NGFCs are compared, as majority of these derive from MGE (based on the previous publication of the authors). These concerns could be addressed by stating explicitly what portion of layer I NGFCs derives from CGE, what portion of GAD65-EGFP and Htr3A-EGFP NGFCs are originated from CGE, or alternatively (most convincing) if a marker of CGE origin was used in the human samples (e.g. posthoc immunolabelling, if available). Additionally, please use "CGE-NGFC" name throughout the manuscript and indicate those recordings that may include MGE-derived NGFCs. It would strengthen the manuscript (but not necessary for this publication) if some of the key findings where tested on MGE-NGFCs because the findings could be extended to a larger population of NGFCs regardless of origin.

4) In Figures 4 and 5, the authors dissect the contribution of K channels to NGFC spiking patterns and convincingly conclude that IKA contributes to the characteristic delayed spiking firing pattern. However, Figure 5I clearly shows that activity-dependent modulation of IA does not underlie STP-SE since STP-SE persists when IA is blocked. Thus, conclusions about the contribution of IA to STP-SE throughout the text and the rationale for identifying channels underlying IA in the context of STP-SE are not convincing. Similarly, while modulation of intrinsic excitability by the blocker AmmTx3 affects synaptic integration (Figure 7D), there is no evidence that this mechanism underlies increased EPSP-spike coupling induced by STP-SE (Figure 7E-I). The claims about how IA contributes to STP-SE require re-evaluation, noting that discounting IA as an expression mechanism for STP-SE does not diminish enthusiasm but seems more consistent with the data shown in Figure 5I. Despite the numerous figures devoted to K channels, in general the conclusions about their specific involvement in STP-SE remain unclear.

5) Further related to the claim that Kv4 function can be activity-dependently regulated, such a statement does not help this manuscript because human samples are derived from epileptic patients where the general activity in the network is usually altered, either by seizure activity or anti-epileptic treatments. Thus, if activity dependent regulation of NGFC excitability is assumed, human and mouse NGFCs would not be comparable. Because, obviously, this question is very complex, I suggest the removal of this section from the Discussion.

6) Albeit the change of EPSCs are small and not significant after AmmTX3 application, the claim that IKA produces EPSP amplification independent of changes in presynaptic release probability is not completely supported by the data. The amplitude of 1st EPSCs and EPSPs changed similarly by approximately 10-20% but the larger variability may prevent significance in the case of EPSCs. Either larger number of voltage clamp recording would be needed, or the data should be interpreted more cautiously.

7) While the data are presented quite transparently, the manuscript suffers somewhat from grammatical inaccuracies and other minor errors (such as reference to the wrong figure panel, unlabeled figure panels, word omissions and duplications). Please go through again very carefully to fix these issues to make the data optimally accessible to the reader. Also, there is a certain tendency for long, overcomplicated sentences.

---

## [Author Response]

Essential revisions:1) The AP peak changed in parallel with the threshold on Figure 3F-G. There is no obvious solution for these two changes because a shift in the threshold implies more efficient sodium channel activation, whereas decrease in the peak of AP implies less efficient sodium channel function. It is important to discuss this paradox, because some technical issues (e.g. offset change) can cause parallel shifts of the two parameters. At least the authors should demonstrate that there are no major changes in the recording conditions by providing data about Rin and Rs parameters. They could also potentially show that the observed effects are specific to AP parameters and that the voltage dependence of independent mechanisms are not changed by the induction protocol, such as the reversal potentials of potassium and sodium (or the more easily measured AMPA and GABA currents).

This is a well taken point and one that is a justified based on the violin plots in Figures 3F/G taken from single mouse and human NGFC examples, respectively. To further investigate the possibility that a technical issue underlies the modulation in action potential parameters observed, we performed additional analyses on all the recorded NGFCs in which STP-SE was elicited. We found that on average no major change in the absolute Vm at the AP peak between baseline and after STP-SE induction (Author response image 1). Unfortunately, the single example violin plots, particularly for the human NGFC, was an extreme one where this value was decreased by approximately 8mV (Author response image 1; lowest orange data point). We have replaced the violin plots with single examples that more closely reflect the mean of the grouped data (Author response image 1). These extra analyses demonstrate that on average the change in the action potential action potential threshold (cf Figure 3H vs. Author response image 1) is greater in magnitude than the changes in absolute Vm at the AP peak discounting the contribution of a recording offset.

**Author response image 1. sa2fig1:** No change in absolute peak of action potentials before and during STP-SE. Black and orange data are from mouse (n = 9) and human(n = 4) NGFCs, respectively.

Furthermore, in some cells in which STP-SE was elicited in the manner outlined in Figure 3 of the manuscript, we continued recording after maximal expression of STP-SE and in all cases a return to baseline of delay to 1^st^ spike, action potential threshold and excitability were observed (single example in Author response image 2; note that the traces depicted here are from the same NGFC used as the single example in Figure 3A). A biphasic change in either offset or access resistance that could underlie this reversal is highly improbable thus further increasing our confidence that such technical issues are not responsible.

**Author response image 2. sa2fig2:** Single example illustrating a reversal in the modulation of AP output following STP-SE induction.

As the reviewers also correctly point out, any increase in intrinsic excitability to a given depolarizing stimulus can be a consequence of alteration in input resistance. We did in fact measure this parameter in a subset of mouse NGFCs before and after STP-SE induction but was not included in the original manuscript. In fact, we saw no significant change in this parameter and therefore we can discount input resistance changes as a contributor to the heightened increase in intrinsic excitability (R_IN_ = 208 ± 53 mΩ and 220 ± 47 mΩ for baseline vs. STP-SE, respectively; n = 11; p = 0.33). These data are now included in the text of the Results section.

Finally, evaluation of the stability of reversal potentials is indeed an elegant means to further address the potential issues raised and we thank the reviewers for this great suggestion. However, in the absence of these additional experiments, we hope that the provided re-analyses and explanation above sufficiently addresses these concerns.

2) It appears in both species that not only the threshold has changed after STP-SE induction, but axonal and somatic components of the AP become more separated (Figure 3FG). This change may indicate a shift in the electrotonic distance of the AP initiation zone from the soma. As the position of AP initiation is crucial for the general excitability, this parameter should be quantified on the available recordings by using the 2nd derivative of the APs. If there is indeed a difference on the whole data set (not only on these two examples), the authors should provide additional discussion how this might contribute to STP-SE.

This is a pertinent point since it is known that acute or chronic bouts of activity can precipitate dynamic structural/functional modifications of the AP initiation site (AIS) resulting in modulation of intrinsic excitability in neurons. As the reviewers point out there appears to be a noticeable inflexion point in the rise phase of the dV/dT AP plots, particularly after STP-SE induction in the mouse (Figure 3F). This is indicative of a shift in the electrotonic distance of the AIS. As suggested, we reanalyzed the data and plotted the second derivative of a representative single AP waveform within baseline and following induction of STP-SE (Author response image 3). In mouse NGFCs, STP-SE was accompanied by either an emergence of two notable peaks in the second derivative plot of the AP rising phase (see NGFC e.g.^1^) or an increase in temporal delay between the two peaks (see NGFC e.g.^2,3^). Indeed, as hypothesized by the reviewers, these data point to an increase in electrotonic distance between APs originating in the AIS and SD compartments. Unfortunately, without complete information concerning the exact location of the AIS (i.e. whether the axon on which it lies directly emanates from the soma or from a primary dendrite), in addition to the length and ion channel composition/functional properties and how this may be modified after STP-SE induction, the consequence of this change in electrotonic distance is difficult to predict. In the absence of this information, one can postulate that an increased electrotonic distance between SD and AIS AP sites would in fact serve to decrease the ability of a dendritic input in eliciting an AP, thus working against our observations of increased E-S coupling. Thus, these changes appear to be counterintuitive and one could argue that additional mechanisms (e.g. increase in somato-dendritic excitability as suggested in the manuscript) are at play to sufficiently overcome the resultant heightened excitability. Furthermore, in a subset of NGFCs in both mouse and human, no obvious change in the second derivative of the AP waveform was noted yet in these cells a similar amount of STP-SE occurred (Author response image 3). Thus, it can be concluded that although changes in electrotonic distances are apparent on a subset of NGFCs it is not essential for the increase in intrinsic excitability observed in STP-SE. We thank the reviewers for this comment and further investigation as part of a separate study would be of great interest since very little is known about the nature of AIS and its potential dynamic modulation in NGFCs, a cell that has one of the most prodigious axonal arborizations in the brain.

**Author response image 3. sa2fig3:** Changes in electrotonic distance of AIS is not a prerequisite for heightened intrinsic excitability observed during STP-SE. Values above 2nd derivative plots denote numbers of NGFCS displaying the changes highlighted. Black and orange data are from mouse (n = 9) and human(n = 4) NGFCs, respectively.

3) The data from mice was obtained from CGE-originated NGFCs as a consequence of using of specific mouse lines. Thus, MGE-derived NGFCs were mostly excluded from the mouse analysis whereas the origin of human NGFCs could not be determined. The authors also discussed that human NGFCs may show larger diversity than NGFCs in rodents but it is not clear how this larger variability contribute to some of the differences between human and mice NGFCs that were observed in this study. Therefore, it would be important to determine the homogeneity of human NGFCs and/or compare with that of mouse NGFCs. This question is particularly relevant when hippocampal NGFCs are compared, as majority of these derive from MGE (based on the previous publication of the authors). These concerns could be addressed by stating explicitly what portion of layer I NGFCs derives from CGE, what portion of GAD65-EGFP and Htr3A-EGFP NGFCs are originated from CGE, or alternatively (most convincing) if a marker of CGE origin was used in the human samples (e.g. posthoc immunolabelling, if available). Additionally, please use "CGE-NGFC" name throughout the manuscript and indicate those recordings that may include MGE-derived NGFCs. It would strengthen the manuscript (but not necessary for this publication) if some of the key findings where tested on MGE-NGFCs because the findings could be extended to a larger population of NGFCs regardless of origin.

As pointed out by the reviewers, based on the use of specific mouse reporter lines the mouse hippocampal and cortical NGFCs examined were exclusively those with a CGE embryonic origin. We now explicitly state this throughout the manuscript.

The dual origin of cortical human NGFCs has recently been revealed by analyses of single cell RNA profiles. MGE-NGFCs in human cortex (like in mouse hippocampus) have soma location in deeper lamina. Of course, a direct confirmation would be definitive, but we are unaware of any approach that has employed immunocytochemical means to directly distinguish CGE-derived interneurons in human. However, with the emergence of recent transcriptomic profiling (single cell RNA-sequencing) of neuronal subtypes, tractable molecular targets are being revealed (e.g. Adenosine Deaminase RNA specific B2; ADARB2; (Hodge et al., 2019. *Nature*. doi: 10.1038/s41586-019-1506-7) that can be probed post-hoc at the protein level as suggested by the reviewers (with the development of appropriate antibodies) to reveal CGE-derived interneurons in the adult human. Nevertheless, in the absence of a direct assessment of embryonic origins, we are confident that based on the laminar position of the human NGFCs examined in the current study are CGE-derived.

We agree that further analyses of MGE-derived NGFCs in both mouse and human will be informative, but we anticipate that the phenomenon described here are indeed universal across the NGFC populations. In fact, we did perform a few recordings from mouse CA1 hippocampal MGE-NGFCs (which have soma predominantly located on the SR-SLM border) and demonstrate that they too can undergo BF like their CGE-counterparts. Furthermore, a recent study demonstrates the ability of hippocampal Ivy cells (a sub-cluster of MGE-NGFCs) can also demonstrate activity-dependent intrinsic plasticity (Krook-Magnuson et al., 2011. *J. Neurosci*. doi: 10.1523/JNEUROSCI.2269-11.2011).

Targeting MGE-NGFCS in human cortex is not trivial and for successful completion would likely require a means to label this subpopulation in live human tissue. Indeed, the development of novel viral strategies in combination with protracted organotypic slice cultures is a plausible technique to allow such targeting of interneuron subtypes in non-human primates/human. Such an approach has already been successfully implemented to label parvalbumin, VIP and LAMP5 subtypes of interneurons in non-human primate and human tissue in a manner accessible for electrophysiological recordings (e.g. Dimidschstein et al., 2019. *BioRxiv* doi:https://doi.org/10.1101/808170l; Mich et al., 2020. *BiorXiv*. doi:https://doi.org/10.1101/555318). Future development of such strategies will undoubtedly result in an increase in our knowledge of the physiological properties of defined IN subtypes, including MGE-NGFCs, embedded in cortical circuits of these higher species. These points have now been added to the Discussion.

4) In Figures 4 and 5, the authors dissect the contribution of K channels to NGFC spiking patterns and convincingly conclude that IKA contributes to the characteristic delayed spiking firing pattern. However, Figure 5I clearly shows that activity-dependent modulation of IA does not underlie STP-SE since STP-SE persists when IA is blocked. Thus, conclusions about the contribution of IA to STP-SE throughout the text and the rationale for identifying channels underlying IA in the context of STP-SE are not convincing. Similarly, while modulation of intrinsic excitability by the blocker AmmTx3 affects synaptic integration (Figure 7D), there is no evidence that this mechanism underlies increased EPSP-spike coupling induced by STP-SE (Figure 7E-I). The claims about how IA contributes to STP-SE require re-evaluation, noting that discounting IA as an expression mechanism for STP-SE does not diminish enthusiasm but seems more consistent with the data shown in Figure 5I. Despite the numerous figures devoted to K channels, in general the conclusions about their specific involvement in STP-SE remain unclear.

These are well taken points and determining a direct link between Kv4 function and increased STP-SE/E-S coupling described requires further interrogation. In response to the lack of occlusion (Figure 5I), although intra-4-AP qualitatively changes AP threshold, delay to spike and excitability in a manner reminiscent of STP-SE, this pharmacological approach does not exactly mimic the extent of these alterations (Figure 4I, 5B). Therefore, the lack of occlusion could be due to incomplete block of I_KA_ and only results in pushing the cell closer to the state of STP-SE as observed. Of course, full genetic ablation (assuming no compensatory mechanisms occur) of Kv4 channels would be one means to provide additional evidence for the role of these channels in the expression of STP-SE. We understand the reviewers’ concern here and as requested toned down our conclusions regarding the role of Kv4 and have highlighted the possible role of additional ion channels. We have explicitly stated throughout the Discussion that additional experimentation is required to unequivocally implicate modulation of this channel in the plasticity phenomenon described here.

5) Further related to the claim that Kv4 function can be activity-dependently regulated, such a statement does not help this manuscript because human samples are derived from epileptic patients where the general activity in the network is usually altered, either by seizure activity or anti-epileptic treatments. Thus, if activity dependent regulation of NGFC excitability is assumed, human and mouse NGFCs would not be comparable. Because, obviously, this question is very complex, I suggest the removal of this section from the Discussion.

As stated in the Introduction (and reiterated in the Materials and methods section) we utilize cortical tissue that is distal to the focal point of the epilepsy. Indeed, most patients underwent cortical resection to access the underlying pathological hippocampus with this former region deemed “normal”. These points are explicitly stated in the Materials and methods section. However, we are cognizant of the pitfalls in using human tissue from epileptic patients that in many instances have undergone aggressive and protracted treatment regimens. Nevertheless, our main take home message of the manuscript is that both mouse and human NGFCs can undergo similar intrinsic plasticity regardless of the existing level of activity at the time of tissue procurement from the two species. We are sensitive to this suggestion regarding conclusions on the mechanistic underpinnings across species and therefore we have amended the Discussion in accordance with this comment.

6) Albeit the change of EPSCs are small and not significant after AmmTX3 application, the claim that IKA produces EPSP amplification independent of changes in presynaptic release probability is not completely supported by the data. The amplitude of 1st EPSCs and EPSPs changed similarly by approximately 10-20% but the larger variability may prevent significance in the case of EPSCs. Either larger number of voltage clamp recording would be needed, or the data should be interpreted more cautiously.

Based on the numbers of experiments performed, our statistical analyses support our interpretation of the results. CGE-NGFCs are known to possess very high expression of functional NMDA receptors when compared to other neuronal subtypes. In fact, we have shown that CGE-NGFCs possess the highest NMDA/AMPA ratio in the interneuron subtypes tested (Chittajallu et al., 2017). By virtue of their longer time course, NMDA EPSPs are well suited to facilitate pronounced summation of incoming excitatory inputs. Indeed, we see a much larger divergence in the change of EPSC and EPSP amplitudes on the second stimulus following AmmTx3 treatment (Figure 7A) and we predict this would be further pronounced with addition stimuli.

7) While the data are presented quite transparently, the manuscript suffers somewhat from grammatical inaccuracies and other minor errors (such as reference to the wrong figure panel, unlabeled figure panels, word omissions and duplications). Please go through again very carefully to fix these issues to make the data optimally accessible to the reader. Also, there is a certain tendency for long, overcomplicated sentences.

We apologize for these stylistic issues and have strived to correct these and other errors accordingly.